# Learning latent causal graphs via mixture oracles

**Bohdan Kivva**
University of Chicago
bkivva@uchicago.edu

**Goutham Rajendran**
University of Chicago
goutham@uchicago.edu

**Pradeep Ravikumar**
Carnegie Mellon University
pradeepr@cs.cmu.edu

**Bryon Aragam**
University of Chicago
bryon@chicagobooth.edu

## Abstract

We study the problem of reconstructing a causal graphical model from data in the presence of latent variables. The main problem of interest is recovering the causal structure over the latent variables while allowing for general, potentially nonlinear dependencies. In many practical problems, the dependence between raw observations (e.g. pixels in an image) is much less relevant than the dependence between certain high-level, latent features (e.g. concepts or objects), and this is the setting of interest. We provide conditions under which both the latent representations and the underlying latent causal model are identifiable by a reduction to a mixture oracle. These results highlight an intriguing connection between the well-studied problem of learning the order of a mixture model and the problem of learning the bipartite structure between observables and unobservables. The proof is constructive, and leads to several algorithms for explicitly reconstructing the full graphical model. We discuss efficient algorithms and provide experiments illustrating the algorithms in practice.

## 1 Introduction

Understanding causal relationships between objects and/or concepts is a core component of human reasoning, and by extension, a core component of artificial intelligence [40, 53]. Causal relationships are robust to perturbations, encode invariances in a system, and enable agents to reason effectively about the effects of their actions in an environment. Broadly speaking, the problem of inferring causal relationships can be broken down into two main steps: 1) The extraction of high-level causal features from raw data, and 2) The inference of causal relationships between these high-level features. From here, one may consider estimating the magnitude of causal effects, the effect of interventions, reasoning about counterfactuals, etc. Our focus in this paper will be the problem of learning causal relationships *between* latent variables, which is closely related to the problem of learning causal representations [63]. This problem should be contrasted with the equally important problem of causal inference in the presence of latent confounders [e.g. 3, 18, 35, 68, 72]; see also Remark 2.1.

Causal graphical models [53, 54] provide a natural framework for this problem, and have long been used to model causal systems with hidden variables [23–26, 58, 59]. It is well-known that in general, without additional assumptions, a causal graphical model given by a directed acyclic graph (DAG) is not identifiable in the presence of latent variables [e.g., 53, 71]. In fact, this is a generic property of nonparametric structural models: Without assumptions, identifiability is impossible, however, given enough structure, identifiability can be rescued. Examples of this phenomenon include linearity [3, 6, 15, 28, 78], independence [1, 10, 78], rank [15, 28], sparsity [6], and graphical constraints [3, 4].

35th Conference on Neural Information Processing Systems (NeurIPS 2021).

In this paper, we consider a general setting for this problem with *discrete* latent variables, while allowing otherwise arbitrary (possibly nonlinear) dependencies. The latent causal graph between the latent variables is also allowed to be arbitrary: No assumptions are placed on the structure of this DAG. We do not assume that the number of hidden variables, their state spaces, or their relationships are known; in fact, we provide explicit conditions under which all of this can be recovered uniquely. To accomplish this, we highlight a crucial reduction between the problem of learning a DAG model over these variables—given access only to the observed data—and learning the parameters of a finite mixture model. This observation leads to new identifiability conditions and algorithms for learning causal models with latent structure.

**Overview**   Our starting point is a simple reduction of the graphical model recovery problem to three modular subproblems:

1. The bipartite graph $\Gamma$ between hidden and observed nodes,
2. The latent distribution $\mathbb{P}(H)$ over the hidden variables $H$, and
3. A directed acyclic graph (DAG) $\Lambda$ over the latent distribution.

From here, the crucial observation is to reduce the recovery problems for $\Gamma$ and $\mathbb{P}(H)$ to the problem of learning a finite mixture over the observed data. The latter is a well-studied problem with many practical algorithms and theoretical guarantees. We do not require parametric assumptions on this mixture, which allows for very general dependencies between the observed and hidden variables. From this mixture model, we extract what is needed to learn the full graph structure.

This perspective leads to a systematic, modular approach for learning the latent causal graph via mixture oracles (see Section 2 for definitions). Ultimately, the application of these ideas requires a practical implementation of this mixture oracle, which is discussed in Section 6.

**Contributions**   More precisely, we make the following contributions:

1. (Section 3) We provide general conditions under which the latent causal model $G$ is identifiable (Theorem 3.2). Surprisingly, these conditions mostly amount to nondegeneracy conditions on the joint distribution. As we show, without these assumptions identifiability breaks down and reconstruction becomes impossible.
2. (Section 4) We carefully analyze the problem of reconstructing $\Gamma$ under progressively weaker assumptions: First, we derive a brute-force algorithm that identifies $\Gamma$ in a general setting (Theorem 4.2), and then under a linear independence condition we derive a polynomial-time algorithm based on tensor decomposition and Jennrich's algorithm (Theorem 4.8).
3. (Section 5) Building on top of the previous step, where we learn the bipartite graph and sizes of the domains of latent variables, we develop an efficient algorithm for learning the latent distribution $\mathbb{P}(H)$ from observed data (Theorem 5.4).
4. (Section 6-7) We implement these algorithms as part of an end-to-end pipeline for learning the full causal graph and illustrate its performance on simulated data.

A prevailing theme throughout is the fact that the hidden variables leave a recognizable "signature" in the observed data through the marginal mixture models induced over subsets of observed variables. By cleverly exploiting these signatures, the number of hidden variables, their states, and their relationships can be recovered exactly.

**Previous work**   Latent variable graphical models have been extensively studied in the literature; as such we focus only on the most closely related work on causal graphical models here. Early work on this problem includes seminal work by Elidan et al. [22], Friedman et al. [27], Martin and VanLehn [47]. More recent work has focused on linear models [3, 28, 68, 78] or known structure [21, 38, 65]. When the structure is not known *a priori*, we find ourselves in the realm of *structure learning*, which is our focus. Less is known regarding structure learning between latent variables for nonlinear models, although there has been recent progress based on nonlinear ICA [36, 50]. For example, [80] proposed CausalVAE, which assumes a linear structural equation model and knowledge of the concept labels for the latent variables, in order to leverage the iVAE model from [36]. By contrast, our results make no linearity assumptions and do not require these additional labels. While this paper was under

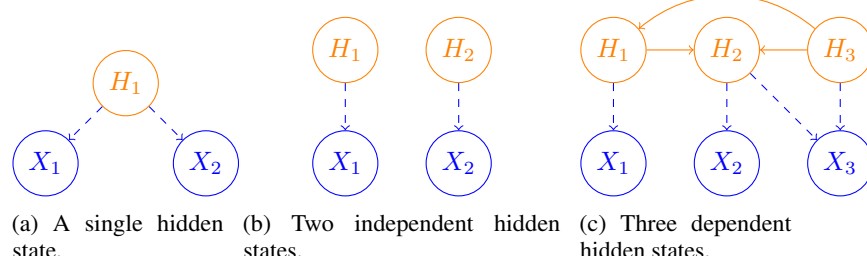

(a) A single hidden state.

(b) Two independent hidden states.

(c) Three dependent hidden states.

Figure 1: Illustration of the basic model. Note that there are no edges between observed variables or edges oriented from observed to hidden. (a) A latent variable model with a single hidden state; i.e. a mixture model. (b)-(c) Two examples of latent variable models with more complicated hidden structure.

review, we were made aware of the recent work [46] that studies a similar problem to ours in a general, nonlinear setting under faithfulness assumptions. It is also worth noting recent progress on learning discrete Boltzmann machines [11, 12], which can be interpreted as an Ising model with a bipartite structure and a single hidden layer—in particular, there is no hidden causal structure. Nevertheless, this line of work shows that learning Boltzmann machines is computationally hard in a precise sense. More broadly, the problem of learning latent structure has been studied in a variety of other applications including latent Dirichlet allocation [8, 9], phylogenetics [51, 64], and hidden Markov models [2, 30].

A prevailing theme in the causal inference literature has been negative results asserting that in the presence of latent variables, causal inference is impossible [20, 32, 61, 62]. Our results do not contradict this important line of work, and instead adopts a more optimistic tone: We show that under reasonable assumptions—essentially that the latent variables are discrete and well-separated—identifiability and exact recovery of latent causal relationships is indeed possible. This optimistic approach is implicit in recent progress on visual relationship detection [52], causal feature learning [14, 43], and interaction modeling [37, 41]. In this spirit, our work provides theoretical grounding for some of these ideas.

**Mixture models and clustering**  While our theoretical results in Sections 3-5 assume access to a mixture oracle (see Definition 2.5), in Section 6 we discuss how this oracle can be implemented in practice. To provide context for these results, we briefly mention related work on learning mixture models from data. Mixture models can be learned under a variety of parametric and nonparametric assumptions. Although much is known about parametric models [e.g. 42], of more interest to us are nonparametric models in which the mixture components are allowed to be flexible, such as mixtures of product distributions [31, 33], grouped observations [60, 75] and general nonparametric mixtures [7, 66]. In each of these cases, a mixture oracle can be implemented without parametric assumptions. In practice, we use clustering algorithms such as $K$-means or hierarchical clustering to implement this oracle. We note also that the specific problem of consistently estimating the order of a mixture model, which will be of particular importance in the sequel, has been the subject of intense scrutiny in the statistics literature [e.g. 16, 19, 39, 45].

**Broader impacts and societal impact**  Latent variable models have numerous practical applications. Many of these applications positively address important social problems, however, these models can certainly be applied nefariously. For example, if the latent variables represent private, protected information, our results imply that this hidden private data can be leaked into publicly released data, which is obviously undesirable. Understanding how to infer unprotected data while safeguarding protected data is an important problem, and our results shed light on when this is and isn't possible.

## 2 Background

Let $G = (V, E)$ be a DAG with $V = (X, H)$, where $X \in \mathbb{R}^n$ denotes the observed part and $H \in \Omega := \Omega_1 \times \cdots \times \Omega_m$ denotes the hidden, or latent, part. Throughout this paper, we assume that each $\Omega_i$ is a discrete space with $|\Omega_i| \geq 2$. We assume further that there are no edges between

observed variables and no edges from observed to hidden variables, and that the distribution of $V$ satisfies the Markov property with respect to $G$ (see the supplement for definitions). Under these assumptions, $G$ decomposes as the union of two subgraphs $G = \Gamma \cup \Lambda$, where $\Gamma$ is a directed, bipartite graph of edges pointing from $H$ to $X$, and $\Lambda$ is a DAG over the latent variables $H$. Similar assumptions have appeared in previous work [3, 46, 78], and although nontrivial, they encapsulate our keen interest in reconstructing the structure $\Lambda$ amongst the latent variables, and captures relevant applications where the relationships between raw observations is less relevant than so-called "causal features" [13, 14]. See Figure 1 for examples.

Throughout this paper, we use standard notation such as $\mathrm{pa}(j)$ for parents, $\mathrm{ch}(j)$ for children, and $\mathrm{ne}(j)$ for neighbors. Given a subset $V' \subset V$, $\mathrm{pa}(V') := \cup_{j \in V'} \mathrm{pa}(j)$ and given a subgraph $G' \subset G$, $\mathrm{pa}_{G'}(V') := \mathrm{pa}(V') \cap G'$, with similar notation for children and neighbors. We let $A \in \{0,1\}^{|X| \times |H|}$ denote the adjacency matrix of $\Gamma$ and denote its columns by $a_j \in \{0,1\}^{|X|}$.

**Remark 2.1.** *Our goal is to learn the hidden variables $H$ and the causal graph between them, defined above by $\Lambda$. To accomplish this, our main result (Theorem 3.2) shows how to identify $(\Gamma, \mathbb{P}(H))$, from which $\Lambda$ can be recovered (see Section 3 for details). It is important to contrast this problem with problems involving latent confounders [e.g. 3, 18, 35, 68, 72], where the goal is to learn the causal graph between the observed variables $X$. In our setting, there are no edges between the observed variables.*

## 2.1 Assumptions

It is well-known that without additional assumptions, the latent variables $H$ cannot be identified from $X$, let alone the DAG $\Lambda$. For example, we can always replace a pair of distinct hidden variables $H_i$ and $H_j$ with a single hidden variable $H_0$ that takes values in $\Omega_i \times \Omega_j$. Similarly, a single latent variable can be split into two or more latent variables. In order to avoid this type of degeneracy, we make the following assumptions:

**Assumption 2.2** (No twins). *For any hidden variables $H_i \neq H_j$ we have $\mathrm{ne}_\Gamma(H_i) \neq \mathrm{ne}_\Gamma(H_j)$.*

**Assumption 2.3** (Maximality). *There is no DAG $G' = ((X, H'), E')$ such that:*

1. *$\mathbb{P}(X, H')$ is Markov with respect to $G'$;*

2. *$G'$ is obtained from $G$ by splitting a hidden variable (equivalently, $G$ is obtained from $G'$ by merging a pair of vertices);*

3. *$G'$ satisfies Assumption 2.2.*

These assumptions are necessary for the recovery of $\Lambda$ in the sense that, without these assumptions, latent variables can be created or destroyed without changing the observed distribution $\mathbb{P}(X)$. Informally, the maximality assumption says that if there are several DAGs that are Markov with respect to the given distribution, we are interested in recovering the most informative among them. Finally, we make a mild assumption on the probabilities, in order to avoid degenerate cases where certain configurations of the latent variables have zero probability:

**Assumption 2.4** (Nondegeneracy). *The distribution over $V = (X, H)$ satisfies:*

(a) *$\mathbb{P}(H = h) > 0$ for all $h \in \Omega_1 \times \ldots \times \Omega_k$.*

(b) *For all $S \subset X$ and $a \neq b$, $\mathbb{P}(S | \mathrm{pa}(S) = a) \neq \mathbb{P}(S | \mathrm{pa}(S) = b)$, where $a$ and $b$ are distinct configurations of $\mathrm{pa}(S)$.*

Without this nondegeneracy condition, $H$ cannot be identified; see the supplement for details.

## 2.2 Mixture oracles

Let $S \subset X$ be a subset of the observed variables. We can always write the marginal distribution $\mathbb{P}(S)$ as

$$\mathbb{P}(S) = \sum_{h \in \Omega} \mathbb{P}(H = h)\mathbb{P}(S \mid H = h). \tag{1}$$

When $S = X$, this can be interpreted as a mixture model with $K := |\Omega|$ components. When $S \subsetneq X$, however, multiple components can "collapse" onto the same component, resulting in a mixture with fewer than $K$ components. Let $k(S)$ denote this number, so that we may define a discrete random variable $Z$ with $k(S)$ states such that for all $j \in [k(S)]$, we have

$$\mathbb{P}(S) = \sum_{j=1}^{k(S)} \underbrace{\mathbb{P}(Z = j)}_{:=\pi(S,j)} \underbrace{\mathbb{P}(S \mid Z = j)}_{:=C(S,j)} = \sum_{j=1}^{k(S)} \pi(S,j)C(S,j). \tag{2}$$

Then $\pi(S, j)$ is the weight of the $j$th mixture component over $S$, and $C(S, j)$ is the corresponding $j$th component. It turns out that these probabilities precisely encode the conditional independence structure of $H$. To make this formal, we define the following oracle:

**Definition 2.5.** *A* mixture oracle *is an oracle that takes $S \subset X$ as input and returns the number of components $k(S)$ as well as the weights $\pi(S, j)$ and components $C(S, j)$ for each $j \in [k(S)]$. This oracle will be denoted by* $\mathsf{MixOracle}(S)$.

Although our theoretical results are couched in the language of this oracle, we provide practical implementation details in Section 6 and experiments to validate our approach in Section 7.

A sufficient condition for the existence of a mixture oracle is that the mixture model over $X$ is identifiable. This is because identifiability implies that the number of components $K$, the weights $\mathbb{P}(Z = j)$, and the mixture components $\mathbb{P}(X \mid Z = j)$ are determined by $\mathbb{P}(X)$. The marginal weights $\pi(S, j)$ and components $C(S, j)$ can then be recovered by simply projecting the full mixture over $X$ onto $S$.

**Remark 2.6.** *In fact, we do not need the full power of* $\mathsf{MixOracle}$. *For our algorithms it is sufficient to have access to $k(S)$ for a sufficiently large family of $S \subset X$, the list of weights $\pi(X, j)$, and a map that relates components in the full mixture over $X$ to the components in the marginal mixtures over each variable $X_i$ (see Section 5 for details).*

Before concluding this section, we note an important consequence of Assumption 2.4 that will be used in the sequel:

**Observation 2.7.** *Under Assumption 2.4, for any $S \subseteq X$*

$$k(S) = \prod_{H_i \in \mathrm{pa}(S)} \dim(H_i) =: \dim(\mathrm{pa}(S)).$$

*Proof.* By the Markov property, $S$ is independent of $H \setminus \mathrm{pa}(S)$. There are $\dim(\mathrm{pa}(S))$ possible assignments to the hidden variables in $\mathrm{pa}(S)$ and by Assumption 2.4, distinct assignments to the hidden variables induce distinct components in the marginal distribution $P(S)$. Hence, by definition, $k(S) = \dim(\mathrm{pa}(S))$. $\qquad\square$

## 3 Recovery of the latent causal graph

We first consider the oracle setting in which we have access to $\mathsf{MixOracle}(S)$.

Observe that the problem of learning $G$ can be reduced to learning $(\Gamma, \mathbb{P}(H))$: Since we can decompose $G$ into a bipartite subgraph $\Gamma$ and a latent subgraph $\Lambda$, it suffices to learn these two components separately. We then further reduce the problem of learning $\Lambda$ to learning the latent distribution $\mathbb{P}(H)$. First, we will show how to reconstruct $\Gamma$ from $\mathsf{MixOracle}(S)$. Then, we will show how to learn the latent distribution $\mathbb{P}(H)$ from $\mathsf{MixOracle}(S)$.

Thus, the problem of learning $G$ is reduced to the mixture oracle:

$$G \to (\Gamma, \mathbb{P}(H)) \to \mathsf{MixOracle}(S).$$

In the sequel, we focus our attention on recovering $(\Gamma, \mathbb{P}(H))$. In order to recover $\mathbb{P}(H)$, we will require the following assumption:

**Assumption 3.1** (Subset condition)**.** *We say that the bipartite graph $\Gamma$ satisfies the subset condition (SSC) if for any pair of distinct hidden variables $H_i, H_j$ the set $\mathrm{ne}_\Gamma(H_i)$ is not a subset of $\mathrm{ne}_\Gamma(H_j)$.*

This assumption is weaker than the common "anchor words" assumption from the topic modeling literature. The latter assumption says that every topic has a word that is unique to this topic, and it is commonly assumed for efficient recovery of latent structure [8, 9].

Under Assumption 3.1, we have the following key result:

**Theorem 3.2.** *Under Assumptions 2.2, 2.3, 2.4, and 3.1, $(\Gamma, \mathbb{P}(H))$ can be reconstructed from $\mathbb{P}(X)$ and $\mathsf{MixOracle}(S)$. Furthermore, if additionally the columns of the bipartite adjacency matrix $A$ are linearly independent, there is an efficient algorithm for this reconstruction.*

The proof is constructive and leads to an efficient algorithm as alluded to in the previous theorem. An overview of the main ideas behind the proof of this result are presented in Sections 4 and 5; the complete proof of this theorem can be found in the supplement.

As presented, Theorem 3.2 leaves two aspects of the problem unresolved: 1) Under what conditions does $\mathsf{MixOracle}(S)$ exist, and 2) How can we identify $\Lambda$ from $\mathbb{P}(H)$? As it turns out, each of these problems is well-studied in previous work, which explains our presentation of Theorem 3.2. For completeness, we address these problems briefly below.

**Existence of** $\mathsf{MixOracle}(S)$    A mixture oracle exists if the mixture model over $X$ is identifiable. As discussed in Section 1, such identifiability results are readily available in the literature. For example, assume that for every $S \subseteq X$, the mixture model (1) comes from any of the following families:

1. a mixture of gaussian distributions [73, 79], or
2. a mixture of Gamma distributions [73], or
3. an exponential family mixture [79], or
4. a mixture of product distributions [74], or
5. a well-separated (i.e. in TV distance) nonparametric mixture [7].

Then $(\Gamma, \mathbb{P}(H))$ is identifiable. The list above is by no means exhaustive, and many other results on identifiability of mixture models are known (e.g., see the survey [48]).

**Identifiability of** $\Lambda$    Once we know $\mathbb{P}(H)$ (e.g. via Theorem 3.2), identifying $\Lambda$ from $\mathbb{P}(H)$ is a well-studied problem with many solutions [53, 71]. For simplicity, it suffices to assume that $\mathbb{P}(H)$ is faithful to $\Lambda$, which implies that $\Lambda$ can be learned up to Markov equivalence. This assumption is *not* necessary, and any number of alternative identifiability assumptions on $\mathbb{P}(H)$ can be plugged in place of faithfulness, for example triangle faithfulness [70], independent noise [56, 67], post-nonlinearity [81], equality of variances [29, 55], etc.

## 4    Learning the bipartite graph

In this section we outline the main ideas behind the recovery of $\Gamma$ in Theorem 3.2. We begin by establishing conditions that ensure $\Gamma$ is identifiable, and then proceed to consider efficient algorithms for its recovery.

### 4.1    Identifiability result

We study a slightly more general setup in which the identifiability of $\Gamma$ depends on how much information we request from the $\mathsf{MixOracle}$. Clearly, we want to rely on $\mathsf{MixOracle}$ as little as possible. As the proofs in the supplement indicate, the only information required for this step are the number of components. Neither the weights nor the components are needed.

**Definition 4.1.** *We say that $\Gamma$ is $t$-recoverable if $\Gamma$ can be uniquely recovered from $X$ and the sequence $(\mathsf{MixOracle}(S) \mid |S| \leq t)$.*

**Theorem 4.2.** *Let $\Gamma$ be the bipartite graph between $X$ and $H$.*

(a) *Assume that $\mathrm{ne}_\Gamma(H_i) \neq \mathrm{ne}_\Gamma(H_j)$ for any $i \neq j$. Then $\Gamma$ and $\dim(H_i)$ are $n$-recoverable.*

(b) *Let $t \geq 3$. Assume that for every $S \subseteq H$ with $|S| \geq 2$ we have*

$$\dim \mathrm{span}\{a_j \mid j \in S\} \geq \frac{2}{t}|S| + 1,$$

*then $\Gamma$ and $\dim(H_i)$ are $t$-recoverable.*

Note that Assumption 3.1 implies the assumption in Theorem 4.2(a). Finally, as in Section 2, we argue that in the absence of additional assumptions, this assumption is in fact necessary:

**Observation 4.3.** *If there is a pair of distinct variables $H_i, H_j \in H$ such that $\mathrm{ne}_\Gamma(H_1) = \mathrm{ne}_\Gamma(H_2)$, then $\Gamma$ is not $n$-recoverable.*

### 4.2 Ideas behind the recovery

In Corollary 4.4 below, we recast Observation 2.7 as an additive identity. This transforms the problem of learning $\Gamma$ into an instance of more general problem that is discussed in the appendix. The results of this section apply to this more general version.

**Corollary 4.4.** *Assume that Assumptions 2.4 hold. For $H_i \in H$ define $w(H_i) = \log(\dim(H_i))$. Then for every set $S \subseteq X$*

$$\log(k(S)) = \sum_{H_i \in \mathrm{pa}(S)} w(H_i). \tag{3}$$

In order to argue about the causal structure of the hidden variables we first need to identify the variables themselves. By Assumption 2.2, every hidden variable leaves a "signature" among the observed variables, which is the set $\mathrm{ne}_\Gamma(H_i)$ of observed variables it affects. In particular, note that $H_i \in \bigcap_{X_s \in \mathrm{ne}_\Gamma(H_i)} \mathrm{pa}(X_s)$, and if there is no $H_j$ with $\mathrm{ne}_\Gamma(H_i) \subset \mathrm{ne}_\Gamma(H_j)$, then $H_i$ is the unique element of the intersection. The lemma above allows us to extract information about the union of parent sets, and we wish to turn it into the information about intersections. This motivates the following definitions.

**Definition 4.5.** *Let $\Gamma$ and $w$ be as above. Define*

$$\mathrm{sne}_\Gamma(S) = \bigcap_{x \in S} \mathrm{ne}_\Gamma(x) \quad and \quad \mathrm{Wsne}_\Gamma(S) = \sum_{v \in \mathrm{sne}_\Gamma(S)} w(v) \tag{4}$$

**Lemma 4.6.** *For a set $S \subseteq X$ we have*

$$\mathrm{Wsne}_\Gamma(S) = \sum_{U \subseteq S, U \neq \emptyset} (-1)^{|U|+1} W_\Gamma(U), \quad where \quad W_\Gamma(S) = \sum_{v \in \mathrm{ne}_\Gamma(S)} w(v). \tag{5}$$

The proof of this lemma is a simple application of the Inclusion-Exclusion principle.

**Remark 4.7.** *The RHS of Eq. (5) only depends on $W$ evaluated on subsets of $S$. Thus, in particular, if $|S| \leq t$ to compute $\mathrm{Wsne}(S)$ it is enough to know* MixOracle *on all sets of size $\leq t$.*

Finally, the values of the function $\mathrm{Wsne}_\Gamma$ can be organized into a tensor, and from here the problem of learning $\Gamma$ can be cast as decomposition problem for this tensor. These proof details are spelled out in the supplement; in the next section we illustrate this procedure for the special case of 3-recovery.

### 4.3 Efficient $3$-recovery

Under a simple additional assumption $\Gamma$ can be recovered efficiently. We are primarily interested in the case $t = 3$. The main idea is to note that a rank-three tensor involving the columns of $A$ can be written in terms of $\mathrm{Wsne}_\Gamma$. We can then apply Jennrich's algorithm [34] to decompose the tensor and recover these columns, which yield $\Gamma$. To see this, let $I = (i_1, i_2, i_3) \subseteq X$ be a triple of indices, and note that

$$\sum_{j \in H} w(j)(a_j)_{i_1}(a_j)_{i_2}(a_j)_{i_3} = \Big( \sum_{j \in H} w(j) a_j \otimes a_j \otimes a_j \Big)_{(i_1, i_2, i_3)} = \mathrm{Wsne}_\Gamma(I). \tag{6}$$

**Theorem 4.8.** *Assume that the columns of $A$ are linearly independent. Then $\Gamma$ and $\dim(H_i)$, for all $i$, are $3$-recoverable in $O(n^3)$ space and $O(n^4)$ time.*

*Proof.* It takes $O(n^3)$ space and $O(n^3)$ time to compute $M_3$ and then Jennrich's algorithm can decompose the tensor in $O(n^3)$ space and $O(n^4)$ time. $\qquad \square$

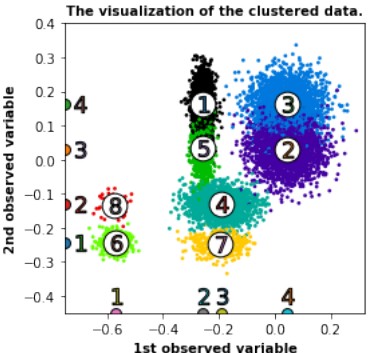
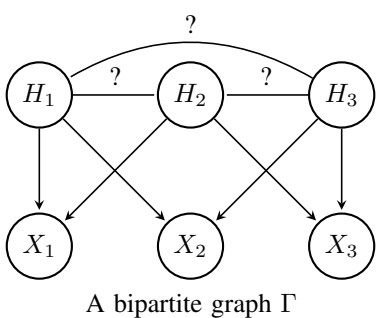

A bipartite graph $\Gamma$

Figure 2: Example of a latent DAG and corresponding mixture distribution

## 5 Learning the latent distribution

In this section we outline the main ideas behind the recovery of $\mathbb{P}(H)$ in Theorem 3.2.

**Remark 5.1.** *Since the variables $H$ are not observed,* MixOracle($S$) *only tells us the set*

$$\{(i, \pi(S, i), C(S, i)) \mid i \in [k(S)]\}.$$

*But the correspondence $\Omega \ni h \leftrightarrow j \in [K]$ between a possible tuple $h$ of values of hidden variables and the corresponding mixture component is unknown.*

Since the values of $H$ are not observed, we may learn this correspondence only up to a relabeling of $\Omega_i$. By definition, the input distribution has $K = |\Omega|$ mixture components over $X$ and $k_i = k(X_i)$ mixture components over $X_i$. Fix any enumeration of these components by $[K]$ and $[k_i]$, respectively. To recover the correspondence $\Omega \ni h \leftrightarrow j \in [K]$, we will need access to the map

$$L : [K] \to [k_1] \times \cdots \times [k_n], \tag{7}$$

defined so that $[L(j)]_i$ equals to the index of the mixture component $C(X, j)$ (marginalized over $X_i$) in the marginal distribution over $X_i$. Crucially, this discussion establishes that $L$ can be computed from a combination of MixOracle($X$) and MixOracle($X_i$) for each $i$.

The map $L$ encodes partial information about the causal structure in $G$. Indeed, if $h_1, h_2 \in \Omega$ are a pair of states of hidden variables $H$ that coincide on pa($X_i$) for some $X_i \in X$, then by the Markov property the components that correspond to $h_1$ and $h_2$ should have the same marginal distribution over $X_i$.

**Example 5.2.** *Consider the DAG on Figure 2. We do not make any assumptions about the causal structure between hidden variables. This DAG has $3$ hidden variables, and we assume that each of them takes values in the set $\{0, 1\}$. Then by Assumption 2.4, every observed variable is a mixture of $4$ components, while the distribution on $X$ is a mixture of $8$ components. Note that the anchor word assumption is violated here, while (SSC) assumption is satisfied. The map $L : [8] \to [4] \times [4] \times [4]$ for an example as in Fig. 2 has form*

| $i$ : | 1 | 2 | 3 | 4 | 5 | 6 | 7 | 8 |
|---|---|---|---|---|---|---|---|---|
| $L(i)$ : | $(2, 4, 3)$, | $(4, 3, 4)$, | $(4, 4, 2)$, | $(3, 2, 4)$, | $(2, 3, 1)$, | $(1, 1, 3)$, | $(3, 1, 2)$, | $(1, 2, 1)$ |

*Our goal is to find the correspondence between $h \in \Omega = \{0, 1\}^3$ and $i \in [8]$. (The projection on the third variable is not shown on Figure 2, so the third coordinate of $L$ cannot be deduced from the plot.)*

We now show that there is an algorithm that exactly recovers $\mathbb{P}(H)$ from the bipartite graph $\Gamma$, the map $L : [K] \to [k_1] \times \cdots \times [k_n]$, and the mixture weights (probabilities) $\{\pi(X, i) \mid i \in [K]\} = \{\mathbb{P}(Z = i) \mid i \in [K]\}$. Each of these inputs can be computed from MixOracle.

**Definition 5.3.** *Let $J$ be an order-$m$ tensor whose $i$-th mode is indexed by values of $H_i$, such that $J(h_1, h_2, \ldots, h_m) = \mathbb{P}(H = h)$. That is, $J$ is the joint probability table of $H$.*

**Theorem 5.4.** *Suppose Assumptions 2.4 and 3.1 hold. Then the correspondence $\Omega \ni h \leftrightarrow C(X, i)$ and the tensor $J(h_1, h_2, \ldots, h_m) = \mathbb{P}(H = (h_1, h_2, \ldots, h_m))$ can be efficiently reconstructed from $L$, $\Gamma$ and $\{\pi(X, i)\}_{i \in [K]}$.*

**Observation 5.5.** *If Assumption 3.1 is violated, then $J$ cannot be reconstructed uniquely. Moreover, in this case $G$ cannot be uniquely identified.*

# 6 Implementation details

The results in Section 3 assume access to the mixture oracle MixOracle($S$). Of course, in practice, learning mixture models is a nontrivial problem. Fortunately, many algorithms exist for approximating this oracle: In our implementation, we used $K$-means. A naïve application of clustering algorithms, however, ignores the significant structure *between* different subsets of observed variables. Thus, we also enforce internal consistency amongst these computations, which makes estimation much more robust in practice. In the remainder of this section, we describe the details of these computations.

**Estimating the number of marginal components**   In order to estimate the number of components in a marginal distribution for a subset $S$ of observed variables with $|S| \leq 3$, we use $K$-means combined with agglomerative clustering to merge nearby cluster centers, and then select the number of components that has the highest silhouette score. Done independently, this step ignores the structure of the global mixture, and is not robust. In order to make learning more robust we observe that the assumptions on the distribution imply the following properties:

- *Divisibility condition:* The number of components we expect to observe over a set $S$ of observed variables is divisible by a number of components we observe on the subset $S' \subset S$ of observed variables (see Obs. 2.7).
- *Structure of means:* Observe that the projections of the means of mixture clusters in the marginal distribution over $S$ are the same as the means of mixture components over variables $S'$ for every $S' \subseteq S$. Hence, if we learn the mixture models over $S$ and $S'$ with the correct numbers of components $k(S)$ and $k(S')$, we expect the projections to be close.

**Example 6.1.** *Suppose we are confident that the number of components in the mixture over $X_1$ is in the set $\{6, 7, 8\}$, over $X_2$ is in $\{4, 5, 6\}$ and the number of components in the mixture over $\{X_1, X_2\}$ is in the set $\{20, 21, 22, 23, 24, 25, 26\}$. Using divisibility condition between $X_1$ and $\{X_1, X_2\}$ we may shrink the set of candidates to $\{21, 24\}$. Next using the divisibility condition for $X_2$ and $\{X_1, X_2\}$ we may determine that the number of components should be $24$.*

With these observations in mind, we use a weighted voting procedure, where every set $S$ votes for the number of components in every superset and every subset based on divisibility or means alignment. We then predict the true number of components by picking the candidate with the most votes.

**Constructing $L$**   In order to estimate $L$ from samples we learn the mixture over the entire set of variables (using K-means and the number of components predicted on the previous step) and over each variable separately (again, using previous step). After this we project the mean of each component to a space over which $X_i$ is defined and pick the closest mean in $L_2$ distance (see Figure 2).

**Reconstructing the latent graphical model**   Once we obtain the joint probability table of $H$, the final piece is to learn the latent DAG $\Lambda$ on $H$. This is a standard problem of learning the causal structure among $m$ discrete variables given samples from their joint distribution. For this a multitude of approaches have been proposed in the literature, for instance the PC algorithm [69] or the GES algorithm [17]. In our experiments, we use the Fast Greedy Equivalence Search [57] with the discrete BIC score, without assuming faithfulness. The final graph $G$ is therefore obtained from $\Gamma$ and $\Lambda$.

# 7 Experiments

We implemented these algorithms in an end-to-end pipeline that inputs observed data and outputs an estimate of the causal graph $G$ and an estimate for the joint probability table $\mathbb{P}(H)$. To test this pipeline, we ran experiments on synthetic data. Full details about these experiments, including a detailed description of the entire pipeline, can be found in Appendix G in the supplement.

**Data generation**   We start with a causal DAG $G$ generated from the Erdös-Rényi model, for different settings of $m, n$ and $|\Omega_i|$. We then generate samples from the probability distribution that corresponds to $G$. We take each mixture component to be a Gaussian distribution with random mean and covariance (we do not force mixture components to be well-separated, aside from constraining the covariances to be small). Additionally, we do not impose restrictions on the weights of the components, which may be very small. As a result, it is common to have highly unbalanced clusters

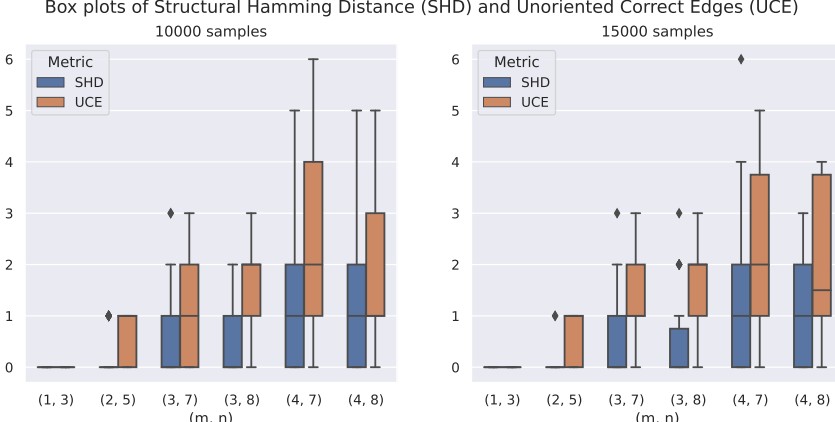

Figure 3: Average Structural Hamming distance for recovery of $G$, where $m = |H|$ and $n = |X|$.

(e.g. we may have less than 30 points in one component and over 1000 in another). Figure 3 reports the results of 600 simulations; 300 each for $N = 10000$ samples and $N = 15000$ samples.

**Results**  To compare how well our model recovers the underlying DAG, we compute the Structural Hamming Distance (SHD) between our estimated DAG and the true DAG. Since GES returns a CPDAG instead of a DAG, we also report the number of correct but unoriented edges in the estimated DAG. The average SHD across different problems sizes ranged from zero to $1.33$. The highest SHD for any single run was $6$. For context, the simulated DAGs had between $3$ and $25$ edges. Note that any errors are entirely due to estimation error in the $K$-means implementation of MixOracle, which we expect can be improved significantly. In the supplement we also report on experiments with much smaller sample size $N = 1000$ (Fig. 7). These results indicate that the proposed pipeline is surprisingly effective at recovering the causal graph.

## 8   Discussion

In this paper, we established general conditions under which the latent causal model $G$ is identifiable (Theorem 3.2). We show that these conditions are essentially necessary, and mostly amount to non-degeneracy conditions on the joint distribution. Under a linear independence condition on columns of the bipartite adjacency matrix of $\Gamma$, we propose a polynomial time algorithm for recovering $\Gamma$ and $\mathbb{P}(H)$. Our algorithms work by reduction to the mixture oracle, which exists whenever the mixture model over $X$, naturally induced by discrete latent variables, is identifiable. Experimental results show effectiveness of our approach. Even though identifiability of mixture models is a long-studied problem, a good mixture oracle implementation is a bottleneck for scalability of our approach. We believe that it may be improved significantly, and consider this as a promising future direction. In this paper, we work under the measurement model that does not allow direct causal relationships between observed variables. We believe that this condition may be relaxed and are eager to explore this direction in future work.

## 9   Acknowledgements

G.R. thanks Aravindan Vijayaraghavan for pointers to useful references. B.K. was partially supported by advisor László Babai's NSF grant CCF 1718902. G.R. was partially supported by NSF grant CCF-1816372. P.R. was supported by NSF IIS-1955532. B.A. was supported by NSF IIS-1956330, NIH R01GM140467, and the Robert H. Topel Faculty Research Fund at the University of Chicago Booth School of Business.

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
