## A  Preliminaries

We say that a distribution $\mathbb{P}(V)$ satisfies the *Markov property* with respect to a DAG $G = (V, E)$ if

$$\mathbb{P}(V) = \prod_{v \in V} \mathbb{P}(v \mid \mathrm{pa}_G(v)). \tag{8}$$

An important consequence of the Markov property is that it allows one to read off conditional independence relations from the graph $G$. More specifically, we have the following [see 53, 71, for details]:

- For each $v \in V$, $v$ is independent of its non-descendants, given its parents.
- For disjoint subsets $V_1, V_2, V_3 \subset V$, if $V_1$ and $V_2$ are $d$-separated given $V_3$ in $G$, then $V_1 \perp\!\!\!\perp V_2 \mid V_3$ in $\mathbb{P}(V)$.

The concept of $d$-separation (see §3.3.1 in [53] or §2.3.4 in [71]) gives rise to a set of independence relations, often denoted by $\mathcal{I}(G)$. The Markov property thus implies that $\mathcal{I}(G) \subset \mathcal{I}(V)$, where $\mathcal{I}(V)$ is the collection of all valid conditional independence relations over $V$. When the reverse inclusion holds, we say that $\mathbb{P}(V)$ is *faithful* to $G$ (also that $G$ is a *perfect map* of $V$). Although the concepts of faithfulness and $d$-separation will not be needed in the sequel, we have included this short discussion for completeness and context (cf. Section 3).

For convenience, we also recall some standard definitions and notation from graphical models.

- The parents of a node $v \in V$ are denoted by $\mathrm{pa}(v) = \{u \in V : (u, v) \in E\}$;
- The children of a node $v \in V$ are denoted by $\mathrm{ch}(v) = \{u \in V : (v, u) \in E\}$;
- The neighborhood of a node $v \in V$ is denoted by $\mathrm{ne}(v) = \mathrm{pa}(v) \cup \mathrm{ch}(v)$.

Throughout the paper and in these appendices, we adopt the convention that $H$ is identified with the indices $[m] = \{1, \ldots, m\}$, and similar $X$ is identified with $[n] = \{1, \ldots, n\}$. In particular, we use $\mathrm{pa}(i)$ and $\mathrm{pa}(H_i)$ interchangeably when the context is clear.

## B  Non-identifiability if Assumption 2.4 is violated

In this appendix we are going to show that Assumptions 2.2 and 2.3 on the graph $G$ are not sufficient for identifiability, and therefore additional assumptions on the distribution of $H$ over $\Omega$ are required as well.

**Definition B.1.** *For distributions $D_1, D_2$, let $D_1 \otimes D_2$ denote the product of the distributions $D_1$ and $D_2$.*

That is, if $X \sim D_1$ and $Y \sim D_2$ are independent, then their joint distribution is $D_1 \otimes D_2$.

The following example illustrates an important case of non-identifiability and motivates the need for Assumption 2.4.

**Example B.2.** *Let $N_0, N_1, N_0', N_1'$ be independent Gaussian distributions with distinct parameters (means and variances). Consider*

$$(X_1, X_2) \sim \frac{1}{2} N_0 \otimes N_0' + \frac{1}{4} N_1 \otimes N_0' + \frac{1}{4} N_1 \otimes N_1' \tag{9}$$

*We claim that $(X_1, X_2)$ is consistent with (i.e., satisfies Markov property with respect to) each of the following three models below. Here, in the model $S_3$ the hidden variable $H_1$ can take three values $\{0, 1, 2\}$, and in models $A$ and $B$, hidden variables take values in $\{0, 1\}$.*

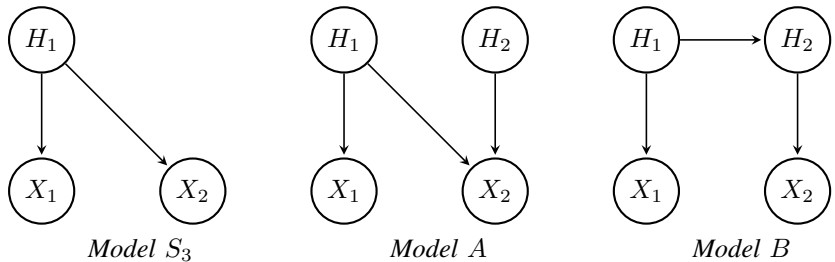

*Model $S_3$*          *Model A*          *Model B*

*Note that all these models satisfy "no-twins" Assumption 2.2 and minimality Assumption 2.3, while Assumptions 2.4 are violated by models A and B.*

1. *Consistency with $S_3$. Let $H_1$ be a random variable that takes values $0, 1, 2$ with probabilities $(1/2, 1/4, 1/4)$. Then*

$$(X_1, X_2) \sim \sum_{j \in \{0,1,2\}} \mathbb{P}(X|H_1 = j) \, \mathbb{P}(H_1 = j), \text{ where}$$

$$\mathbb{P}(X|H_1 = 0) = N_0 \otimes N_0', \quad \mathbb{P}(X|H_1 = 1) = N_1 \otimes N_0', \quad \mathbb{P}(X|H_1 = 2) = N_1 \otimes N_1'$$

2. *Consistency with A. Let $H_1$ and $H_2$ be i.i.d random variables that take values $0, 1$ with probabilities $(1/2, 1/2)$. Then*

$$(X_1, X_2) \sim \sum_{i \in \{0,1\}} \sum_{j \in \{0,1\}} \mathbb{P}(X|H_1 = i, H_2 = j) \, \mathbb{P}(H_1 = i) \, \mathbb{P}(H_2 = j), \text{ where}$$

$$\mathbb{P}(X|H_1 = 0) = N_0 \otimes N_0', \quad \mathbb{P}(X|H_1 = 1, H_2 = 0) = N_1 \otimes N_0',$$
$$\mathbb{P}(X|H_1 = 1, H_2 = 1) = N_1 \otimes N_1'$$

3. *Consistency with B. Let $H_1$ be a random variable that takes values $0, 1$ with probabilities $(1/2, 1/2)$. Let $H_2$ be a dependent random variable that takes values $0, 1$ with probabilities $(1, 0)$, if $H_1 = 0$, and with probabilities $(1/2, 1/2)$, if $H_1 = 1$.*

$$(X_1, X_2) \sim \sum_{i \in \{0,1\}} \sum_{j \in \{0,1\}} \mathbb{P}(X_1|H_1 = i) \, \mathbb{P}(X_2|H_2 = j) \, \mathbb{P}(H_1 = i) \, \mathbb{P}(H_2 = j|H_1 = i),$$

   *where*
$$\mathbb{P}(X_1|H_1 = 0) = N_0, \quad \mathbb{P}(X_1|H_1 = 1) = N_1,$$
$$\mathbb{P}(X_2|H_2 = 0) = N_0', \quad \mathbb{P}(X_2|H_2 = 1) = N_1'$$

**Remark B.3.** *Observe that among the models $A, B$ and $S_3$, only $S_3$ satisfies Assumption 2.4. Observe that the model A satisfies part (a), but not (b), and the model B satisfies part (b), but not (a), of Assumption 2.4. This shows that only one of these assumptions is still not sufficient for identifiability of a latent causal model.*

## C   Reconstructing bipartite part $\Gamma$. Proofs for Sections 4

Recall that (cf. Section 4.2), that for $w(H_i) = \log(\dim(H_i))$ and every subset $S \subseteq X$ the parameters of the latent DAG satisfy

$$\log(k(S)) = \sum_{H_i \in \mathrm{pa}(S)} w(H_i). \tag{10}$$

Recall also the definitions of sne and Wsne in (4), reproduced here for ease of reference:

$$\mathrm{sne}_\Gamma(S) = \bigcap_{x \in S} \mathrm{ne}_\Gamma(x) \quad \text{and} \quad \mathrm{Wsne}_\Gamma(S) = \sum_{v \in \mathrm{sne}_\Gamma(S)} w(v).$$

## C.1 Learning a bipartite graph with a hidden part from an additive score

We start our discussion of the proof of results in Section 4 by reducing learning of the causal graph $\Gamma$ to a more general learning problem.

Let $\Gamma = (X \cup H, E)$ be a (not necessarily directed) bipartite graph on parts $X$ and $H$, and let $w : H \to (0, \infty)$ be an arbitrary function that defines weights of variables in $H$.

Recall that for a weight function $w$ and subset $S \subseteq X$ we define

$$W_\Gamma(S) = \sum_{v \in \mathrm{ne}_\Gamma(S)} w(v) \tag{11}$$

**Problem C.1.** *Assume that the vertices in $H$ and the weight function $w$ are unknown.*

*Input: Values $(W_\Gamma(S) \mid S \in \mathcal{F})$ indexed by a family of known subsets $\mathcal{F} \subseteq 2^X$*

*Goal: Reconstruct the number of unknown vertices $H$, the graph $\Gamma$ between $H$ and $X$ (up to an isomorphism), and the weight function $w$ from the input.*

Whether it is possible to reconstruct $\Gamma$ and $w$ from the input may depend on the family $\mathcal{F}$ or some additional assumptions about the structure of the graph $\Gamma$. To account for weights $w$, we slightly modify Definition 4.1 as follows:

**Definition C.2.** *We say that $(\Gamma, w)$ is $\mathcal{F}$-recoverable if $(\Gamma, w)$ can be uniquely recovered from $X$ and the sequence $(W_\Gamma(S) \mid S \in \mathcal{F})$.*

In the sequel, we use this modified definition.

The most natural regime is when $\mathcal{F}$ contains the sets whose size is bounded:

**Definition C.3.** *We say that $(\Gamma, w)$ is $t$-recoverable if $(\Gamma, w)$ is $\binom{X}{\leq t}$-recoverable, where $\binom{X}{\leq t}$ denotes the collection of subsets of $X$ of size at most $t$.*

## C.2 Reconstructing $\Gamma$ with full information about $W$

In this section we study Problem C.1, when full information about $W_\Gamma(\cdot)$ is provided, i.e. $\mathcal{F} = 2^X$.

Although the algorithm considered here will have exponential in $|X|$ runtime, it sheds light on the minimal theoretical assumptions we need for proving identifiability of $\Gamma$. We will consider more efficient algorithms in later sections.

We start by proving Observation 4.3, which notes that if $\mathrm{ne}_\Gamma(H_i) = \mathrm{ne}_\Gamma(H_j)$ for $H_i \neq H_j$, then $(\Gamma, w)$ is not $2^X$-recoverable.

*Proof of Observation 4.3.* Consider the graph $\Gamma'$ obtained from $\Gamma$ by replacing $H_1$ and $H_2$ with a single variable $H^*$ and by connecting $H^*$ by an edge to all vertices in $X$ that are adjacent with $H_1$ or $H_2$ in $\Gamma$. Define $w(H^*) = w(H_1) + w(H_2)$. Then $W_\Gamma(S) = W_{\Gamma'}(S)$ for any $S \subseteq X$. $\qquad\square$

**Corollary C.4.** *Let $\mathcal{F} \subseteq 2^X$. If there is a pair of distinct variables $H_i, H_j \in H$ such that $\mathrm{ne}_\Gamma(H_1) = \mathrm{ne}_\Gamma(H_2)$, then $(\Gamma, w)$ is not $\mathcal{F}$-recoverable.*

We now prove that in the case $\mathcal{F} = 2^X$, this is the only obstacle. We start by showing that certain neighborhoods of hidden variables can be identified using $\mathrm{Wsne}(\cdot)$.

As explained in Section 4.2, in the case when $\mathrm{ne}(H_i) \not\subset \mathrm{ne}(H_j)$ for all $H_j$, we expect $\mathrm{Wsne}(\cdot)$ to have a clear "signature" of $H_i$. We make this intuition precise in the definition and lemma that follows.

**Definition C.5.** *We say that a set $S$ of observed variables $X$ is a* maximal neighborhood block *if $\mathrm{Wsne}(S) \neq 0$, but for any superset $S'$ of $S$ we have $\mathrm{Wsne}(S') = 0$.*

**Lemma C.6.** *A set $S \subseteq X$ is a maximal neighborhood block if and only if there exists a hidden vertex $H_i \in H$ such that $\mathrm{ne}_\Gamma(H_i) = S$ and for any other $H_j \in H$ we have $S \not\subseteq \mathrm{ne}_\Gamma(H_j)$.*

*Proof.* Assume that $S \subseteq X$ is a maximal neighborhood block. Since $\mathrm{Wsne}(S) > 0$ the set of common neighbours $\mathrm{sne}_\Gamma(S)$ is non-empty. If $\mathrm{sne}_\Gamma(S)$ contains a hidden vertex $H_j$ that is connected

to a vertex $x \notin S$ then, $H_j \in \mathrm{sne}_\Gamma(S \cup \{x\})$, and $\mathrm{Wsne}_\Gamma(S \cup \{x\}) \geq w(H_j) > 0$ which contradicts the assumption that $S$ is a maximal neighborhood block. Therefore, for every $H_j$ in $\mathrm{sne}_\Gamma(S)$, we have $\mathrm{ne}_\Gamma(H_j) \subset S$. Therefore, there exists a variable $H_i$ such that $\mathrm{ne}_\Gamma(H_i) = S$ and for any other $H_j \in H$ we have $S \not\subseteq \mathrm{ne}_\Gamma(H_j)$.

The opposite implication can be verified in a similar way. $\qquad\square$

**Theorem C.7** (Theorem 4.2, part (a)). *Let $\Gamma$ be a bipartite graph with parts $X$ and $H$. Assume that no pair of vertices in $H$ has the same set of neighbours (in $X$). Then $\Gamma$ is $2^X$-recoverable.*

*Proof.* We prove the claim of the theorem by induction on $|H|$. The statement for the base case $|H| = 0$ immediately follows from the fact that $W(S) = 0$ for all $v \in X$ if and only if $|H| = 0$ since $w(\cdot) > 0$. Assume that we proved the claim for all $\Gamma$ with $|H| = t$ that satisfy the assumptions of the theorem. Let $\Gamma$ be a graph with $|H| = t + 1$ that satisfies the assumptions of the theorem.

Using Lemma 4.6, compute values $\mathrm{Wsne}_\Gamma(S)$ for every $S \subseteq X$. Using values of $\mathrm{Wsne}(\cdot)$ we can find a maximal neighborhood block $Y \subseteq X$. By Lemma C.6, there exists a hidden vertex $H_i$ such that $\{H_i\} = \mathrm{sne}_\Gamma(Y)$. Note that $w(H_i) = \mathrm{Wsne}(Y)$.

Denote by $\Gamma'$ the graph obtained from $\Gamma$ by deleting $H_i$.

Now we verify that $\Gamma'$ satisfies the assumptions of the theorem. There is nothing to check if the set of hidden vertices of $\Gamma'$ is empty. Assume that $\Gamma'$ has a non-empty set of hidden vertices. First, note that all hidden vertices in $\Gamma'$ still have distinct sets of neighbors. Second, note that (cf. (11)) $W_{\Gamma'}(S) = W_\Gamma(S)$ if $S \cap Y = \emptyset$ (i.e. $H_i \notin \mathrm{ne}_\Gamma(S)$), and

$$W_{\Gamma'}(S) = W_\Gamma(S) - w(H_i) = W_\Gamma(S) - \mathrm{Wsne}_\Gamma(Y)$$

if $S \cap Y$ is not empty. Thus, we can compute $W_{\Gamma'}$ from the values of $W_\Gamma$.

By the induction hypothesis $(\Gamma', w|_{\Gamma'})$ is uniquely recoverable from $W_{\Gamma'}(S)$. Let $\Gamma^*$ be the graph obtained from $\Gamma'$ by adding a new variable $H_Y$ of weight $\mathrm{Wsne}_\Gamma(Y)$ and edges between $H_Y$ and $Y$. Then $\Gamma^*$ is isomorphic to $\Gamma$, and so $\Gamma$ is $2^X$-recoverable. $\qquad\square$

## C.3 Efficient $t$-recovery of $\Gamma$ for $t \geq 3$

The approach proposed in Appendix C.2 is exponential in the number of observed variables in the worst case, since we need to compute the scores of all subsets of $X$. In this section, we show that with a mild additional assumption, there is an efficient algorithm to learn the bipartite graph between hidden and observed variables.

As before, let $\Gamma = (X \cup H, E)$ be the bipartite graph between hidden and observed variables.

Recall, that we defined $A$ to be the $|X| \times |H|$ adjacency matrix of $\Gamma$ (with $0, 1$ entries) and $a_i$ to denote the $i$-th column of $A$.

For a sequence of indices $I = (i_1, i_2, \ldots, i_t) \subseteq [n]$ define

$$\mathrm{Wsne}_\Gamma(I) = \sum_{j \in H} w(j) \underbrace{(a_j)_{i_1}(a_j)_{i_2} \ldots (a_j)_{i_t}}_{t} = \left( \sum_{j \in H} w(j) \underbrace{a_j \otimes a_j \otimes \ldots \otimes a_j}_{t} \right)_{(I)}. \qquad (12)$$

Recall, that as pointed out in Remark 4.7, for any $S \subseteq X$ with $|S| \leq t$ the value $\mathrm{Wsne}_\Gamma(S)$ can be computed from the $\{W_\Gamma(S) \mid S \subseteq X, |S| \leq t\}$ using Lemma 4.6. Therefore, we can make the following observation.

**Observation C.8.** *All entries of the the tensor $M_t = \sum_{j \in H} w(j) (\underbrace{a_j \otimes a_j \otimes \ldots \otimes a_j}_{t})$ can be computed as $M_t(I) = \mathrm{Wsne}_\Gamma(I)$ in $O(2^t n^t)$ time and space assuming access to $\{W_\Gamma(S) \mid S \subseteq X, |S| \leq t\}$.*

For fixed $t$ this is a poly-time computation. Furthermore, in the settings we consider in Secrion 4 the values of $W_\Gamma$ can be computed from MixOracle using Observation 2.7.

Now we want to recover the vectors $a_j$ from $M_t$. Since $a_j$ are the columns of the adjacency matrix of $\Gamma$ this is equivalent to recovering the adjacency matrix of $\Gamma$ or $\Gamma$ itself up to an isomorphism.

**Definition C.9.** *For an order-t tensor $M_t$ its rank is defined as the smallest $r$ such that $M_t$ can be written as*

$$M_t = \sum_{j=1}^{r} c_j \bigotimes_{i=1}^{t} x_j^{(i)}. \tag{13}$$

*Such decomposition of $M$ with precisely $r$ components is called a minimum rank decomposition or a CP-decomposition.*

**Lemma C.10.** *If the decomposition*

$$M_t = \sum_{j \in H} w(j) \underbrace{a_j \otimes a_j \otimes \ldots \otimes a_j}_{t}$$

*is the unique minimum rank decomposition, then $(\Gamma, w)$ is $t$-recoverable.*

*Proof.* In order to recover $\Gamma$ and $w$ we compute $M_t$ using $\{W_\Gamma(S) \mid S \subseteq X, |S| \leq t\}$. Then $a_j$ and $w(j)$ can be uniquely (up to permutation) identified from minimum rank decomposition of $M$. $\quad\square$

The following simplified version of Kruskal's condition was proposed by Lovitz and Petrov.

**Theorem C.11** ([44, Theorem 2]). *Let $m \geq 2$ and $t \geq 3$ be integers. Let $V = V_1 \otimes V_2 \otimes \ldots \otimes V_t$ be a multipartite vector space over a field $\mathbb{F}$ and let*

$$\{x_j^{(1)} \otimes x_j^{(2)} \otimes \ldots \otimes x_j^{(t)} \mid j \in [m]\} \subset V\}$$

*be a set of $m$ rank-1 (product) tensors. For a subset $S \subseteq [m]$ with $|S| \geq 2$ and $j \in [t]$ define*

$$d_i(S) = \dim \operatorname{span}\{x_j^{(i)} \mid j \in S\}.$$

*If $2|S| \leq \sum_{i=1}^{t}(d_i(S) - 1) + 1$ for every such $S$, then*

$$\sum_{j \in [m]} x_j^{(1)} \otimes x_j^{(2)} \otimes \ldots \otimes x_j^{(t)}$$

*constitutes a unique minimal rank decomposition.*

In our settings the sufficient condition for having the unique minimal rank decomposition takes the following form.

**Corollary C.12.** *Assume that for every $S \subseteq H$ with $|S| \geq 2$ we have*

$$\dim \operatorname{span}\{a_j \mid j \in S\} \geq \frac{2}{t}|S| + 1,$$

*then the decomposition $M_t = \sum_{j \in H} w(j) \underbrace{a_j \otimes a_j \otimes \ldots \otimes a_j}_{t}$ is the unique minimum rank decomposition and so $(\Gamma, w)$ is $t$-recoverable.*

*Proof.* Take $\mathbb{F} = \mathbb{R}$, then the result follows from C.11 for $x_j^{(1)} = w(j)a_j$ and $x_j^{(i)} = a_j$. $\quad\square$

*Proof of Theorem 4.2 part (b).* Follows by combining Corollary C.12 and Lemma C.10. $\quad\square$

Learning the components of the minimum rank decomposition is a very well-studied problem for which a variety of algorithms have been proposed in the literature (see the survey [76] or the book [49]). We can use Jennrich's algorithm [34] (see also [49, 76] and the references therein) as an efficient algorithm with guarantees:

**Theorem C.13** (Jennrich's algorithm [34]). *Assume that the components of the tensor $\mathcal{T} = \sum_{i=1}^{r} a_i \otimes b_i \otimes c_i$ satisfy the following conditions. The vectors $\{a_i \mid i \in [r]\}$ are linearly independent, the vectors $\{b_i \mid i \in [r]\}$ are linearly independent, and no pair of vectors $c_i, c_j$ is linearly dependent for $i \neq j$. Then the components of the tensor can be uniquely recovered in $O(n^3)$ space and $O(n^4)$ time.*

**Remark C.14.** *Note that if all vectors $a_i$ are linearly independent, then the assumptions of Corollary C.12 are satisfied.*

**Remark C.15.** *A similar problem for $t$-recovery (for weighted hypergraphs) arose in a completely different context [5]. While in both papers the problem is reduced to recovering the minimum rank decomposition of a carefully constructed tensor, we give better recovery guarantees for this problem by using more recent uniqueness guarantees [44].*

# D    Reconstruction of the probability distribution on $H$. Proofs for Section 5

In this section we discuss how one may reconstruct the hidden probability distribution on $\mathbb{P}(H)$ from

- the bipartite graph $\Gamma$, and
- the function $L : [K] \to [k_1] \times \cdots \times [k_n]$, and
- the mixture weights (probabilities) $\{\pi(X, i) \mid i \in [k(X)]\} = \{\mathbb{P}(Z = i) \mid i \in [k(X)]\}$

## D.1    A key lemma

Below we formulate the key lemma that allows us to relate the structure present in the map $L$ with the causal structure in $G$.

Given a state $H = (h_1, \dots, h_m)$ and its corresponding component $P(X \mid H_1 = h_1, \dots, H_m = h_m)$, we want to identify the components $P(X \mid H_1 = h'_1, H_2 = h_2, \dots, H_m = h_m)$ that result from changing just the first hidden variable while keeping every other hidden variable fixed. The next lemma says that we can identify such components by looking into the distribution of the observed variables that are not children of $H_1$.

**Lemma D.1.** *Let $H_i$ be a hidden variable and let $C(X \setminus \mathrm{ne}_\Gamma(H_i), j)$ be an arbitrary mixture component observed in a marginal mixture distribution over the variables in $X \setminus \mathrm{ne}_\Gamma(H_i)$. Let $C(j_1), C(j_2), \dots C(j_t)$ be all the mixture components in the distribution of $X$ whose marginal distribution over $X \setminus \mathrm{ne}_\Gamma(H_i)$ is equal to $C(X \setminus \mathrm{ne}_\Gamma(H_i), j)$. In other words, $L(j_s)_i = j$ for all $s \in [t]$. Then $t = \dim(H_i)$ and every $C(j_s)$ for $s \in [t]$ corresponds to a distinct value of $H_i$.*

*Proof.* Observe that Assumption 3.1 implies that $\mathrm{ne}_\Gamma(X \setminus \mathrm{ne}_\Gamma(H_i)) = H \setminus \{H_i\}$. Therefore, by Assumption 2.4(b), $p(X \setminus \mathrm{ne}_\Gamma(H_i) \mid H = h_1) \sim p(X \setminus \mathrm{ne}_\Gamma(H_i) \mid H = h_2)$, if and only if $h_1$ and $h_2$ differ only in the value of $H_i$. $\square$

## D.2    Examples

Prior to presenting our algorithm in full generality we show how it works on Example 5.2 from Section 5. The basic idea is the following: We start by arbitrarily assigning a component $C(X, i)$—and hence its corresponding probability $\pi(X, i)$ to some hidden state $h^* = (h_1, \dots, h_m)$. This assignment amounts to declaring $\mathbb{P}(H_1 = h_1, \dots, H_m = h_m) = \pi(X, i)$ and $\mathbb{P}(X \mid H_1 = h_1, \dots, H_m = h_m) = C(X, i)$. The choice of initial state $h^*$ here is immaterial; this can be done without loss of generality since the values of the hidden variables can be relabeled without changing anything. From here we proceed inductively by considering hidden states that differ from the previously identified states by in exactly one coordinate. In the example below, we start with $h^* = (0, \dots, 0)$ and then use this as a base case to identify $h^* + e_i$ for each $i = 1, \dots, m$, where

$$(e_i)_j = \begin{cases} 1 & i = j \\ 0 & i \neq j. \end{cases}$$

Note that $h^*$ and $e_i$ differ in exactly one coordinate. We then repeat this process until all states have been exhausted. The following example illustrates the procedure and explains how Lemma D.1 helps to resolve the ambiguity regarding the assignment of components to hidden states in each step.

**Example D.2.** *Consider the DAG $G$ in Fig. 4. It has $3$ hidden variables, each of which takes values in $\{0, 1\}$. By Assumption 2.4 every observed variable is a mixture of $4$ components, while the*

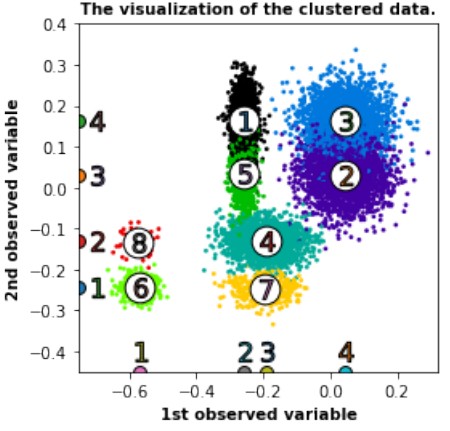 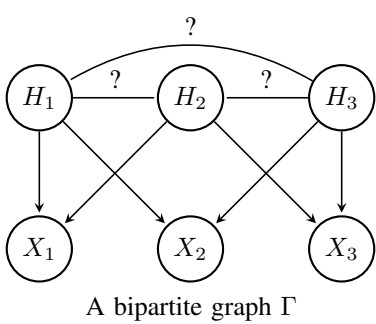

A bipartite graph $\Gamma$

Figure 4: Example of $\mathbb{P}(H)$ learning

*distribution on $X$ is a mixture of $8$ components. Note that the anchor word assumption is violated here, while SSC (Assumption 3.1) is satisfied. The map $L : [8] \to [4] \times [4] \times [4]$ can be written as:*

| $i :$ | 1 | 2 | 3 | 4 | 5 | 6 | 7 | 8 |
|---|---|---|---|---|---|---|---|---|
| $L(i) :$ | $(2,4,3),$ | $(4,3,4),$ | $(4,4,2),$ | $(3,2,4),$ | $(2,3,1),$ | $(1,1,3),$ | $(3,1,2),$ | $(1,2,1)$ |

*We want to find the correspondence between $h \in \Omega = \{0,1\}^3$ and $i \in [8]$.*

*We start by picking an arbitrary component, say 1, and assign it to $(H_1, H_2, H_3) = (0,0,0)$. Next, we make use of Lemma D.1. Since we know $\Gamma$, we know $\mathrm{ch}(H_i)$ for each $i$. In particular, for the hidden variable $H_1$, we know $\mathrm{ch}(H_1) = \{X_1, X_2\}$. This implies that if $H_2, H_3$ are fixed while $H_1$ changes its value, then the component of $X_3$ is unchanged. It follows that the third coordinate of $L$ is also unchanged. This gives us a way to pair up the components that have the same third coordinate $L(i)_3$; the pairs are $(1,6)$, $(2,4)$, $(3,7)$ and $(5,8)$. By our previous observation, these pairs are in one-to-one correspondence with unique states of $(H_1, H_2) = (h_1, h_2)$, and each pair identifies the pair of components $(P(X \mid H_1 = 0, H_2 = h_2, H_3 = h_3), P(X \mid H_1 = 1, H_2 = h_2, H_3 = h_3))$. Note that at this stage, there is still ambiguity as to which coordinate of each pair corresponds to which component.*

*Similarly, we can pair up the components that correspond to assignments of hidden variables that differ only in the value of $H_2$. The pairs are $(1,3)$, $(2,5)$, $(4,8)$ and $(6,7)$. Finally, for $H_3$ the pairs are $(1,5)$, $(2,3)$, $(4,7)$ and $(6,8)$.*

*Since component 1 is assigned to $(H_1, H_2, H_3) = (0,0,0)$ we can deduce that*

| $(H_1, H_2, H_3) :$ | $(0,0,0),$ | $(1,0,0),$ | $(0,1,0),$ | $(1,1,0),$ | $(0,0,1),$ | $(1,0,1),$ | $(0,1,1),$ | $(1,1,1)$ |
|---|---|---|---|---|---|---|---|---|
| $comp.\# :$ | 1 | 6 | 3 | ? | 5 | ? | ? | ? |

*Assume that we know which components correspond to the hidden variable state $(H_1, H_2, H_3) = (h_1, h_2', h_3)$ and $(H_1, H_2, H_3) = (h_1', h_2, h_3)$, with $h_1 \neq h_1'$ and $h_2 \neq h_2'$. Then we can use the information above to deduce which components correspond to the hidden state $(h_1', h_2', h_3)$ since it differs from them in just 1 position. Hence, we can deduce*

| $(H_1, H_2, H_3) :$ | $(0,0,0),$ | $(1,0,0),$ | $(0,1,0),$ | $(1,1,0),$ | $(0,0,1),$ | $(1,0,1),$ | $(0,1,1),$ | $(1,1,1)$ |
|---|---|---|---|---|---|---|---|---|
| $comp.\# :$ | 1 | 6 | 3 | 7 | 5 | 8 | 2 | ? |

*Note that since $(1,1,1)$ differs from the four states identified in the first step in two entries, this has not been determined yet. However, repeating this argument a third time we can deduce that component 4 corresponds to $(H_1, H_2, H_3) = (1,1,1)$.*

To illustrate how algorithm works in the case of non-binary latent variables we provide one more example.

**Example D.3.** *Assume that $\mathbb{P}(V)$ is Markov with respect to the DAG $G$ in Figure 5 where we make no assumption about causal relation between $H_1$ and $H_2$. Assume that $\dim(H_1) = \dim(H_2) = 3$.*

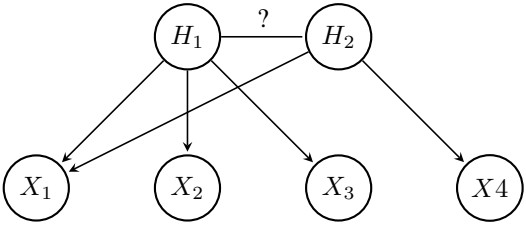

Figure 5: A bipartite graph $\Gamma$ in Example D.3

*Suppose that the map $L : [9] \to [9] \times [3] \times [3] \times [3]$ is given by:*

$$
\begin{array}{cccccc}
i : & 1 & 2 & 3 & 4 & 5 \\
L(i) : & (1,2,1,3), & (3,3,3,1), & (4,1,2,2), & (2,2,1,1), & (7,2,1,2), \\
i : & 6 & 7 & 8 & 9 \\
L(i) : & (5,1,2,1), & (9,1,2,3), & (8,3,3,3) & (6,3,3,2)
\end{array}
$$

*We want to find the correspondence between $h \in \Omega = \{0,1,2\}^2$ and $i \in [9]$.*

*As in the previous example, in order to see which components correspond to the states of latent variables where $H_2$ is fixed and $H_1$ takes all values in $\{0,1,2\}$ we group together the components that have the same value of $L$ on $X \setminus \mathrm{ch}(H_1) = \{X_4\}$. We get the following groups $(1,7,8)$, $(2,4,6)$ and $(3,5,9)$.*

*Similarly, by comparing the values of $L$ on $X \setminus \mathrm{ch}(H_2) = \{X_2, X_3\}$ we get that the following groups correspond to a fixed value of $H_1$, while $H_2$ vary: $(1,4,5)$, $(2,8,9)$ and $(3,6,7)$.*

*Since values of $H_i$ are determined up to relabeling we can arbitrarily assign a component, say 1, to $(H_1 = 0, H_2 = 0)$. Now, using Lemma D.1, we know that components that correspond to $(H_1 = 1, H_2 = 0)$ and $(H_1 = 2, H_2 = 0)$ are 7 and 8, and again because values of $H_i$ can be relabeled, at this point the choice is arbitrary. Using the similar argument for $H_2$, we can deduce the following correspondence:*

$$
\begin{array}{cccccccccc}
(H_1, H_2) : & (0,0), & (1,0), & (2,0) & (0,1), & (1,1), & (2,1), & (0,2), & (1,2), & (2,2) \\
comp.\# : & 1 & 7 & 8 & 4 & ? & ? & 5 & ? & ?
\end{array}
$$

*At this point the labeling of the values of hidden variables is fixed. Now let us consider an index of hamming weight 2, say $(1,1)$. We know that the component, that corresponds to this state of latent variables, differs from the component 4, that corresponds to $(0,1)$, only due to the change of $H_1$. Hence, the component that corresponds to $(1,1)$ is in the set $\{2,4,6\}$. At the same time, we know that it differs from the component 7 that corresponds to $(1,0)$ only due to the change of $H_2$. Hence, the desired component is in the set $\{3,6,7\}$. By taking the intersection of sets $\{2,4,6\}$ and $\{3,6,7\}$ we deduce that the value that corresponds to $(1,1)$ is 6. Similarly we can determine the rest of the values.*

$$
\begin{array}{cccccccccc}
(H_1, H_2) : & (0,0), & (1,0), & (2,0) & (0,1), & (1,1), & (2,1), & (0,2), & (1,2), & (2,2) \\
comp.\# : & 1 & 7 & 8 & 4 & 6 & 2 & 5 & 3 & 9
\end{array}
$$

### D.3 Proof of Theorem 5.4

The algorithm described in the previous examples can be used to prove Theorem 5.4. For this, we present a general algorithm to recover the correspondence $\Omega \ni h \leftrightarrow i \in [K]$ using Lemma D.1.

*Proof of Theorem 5.4.* Without loss of generality, we may assume that $H_i$ takes values from $\Omega_i = \{0, 1, \ldots, \dim(H_i) - 1\}$ for every $i$.

Recall that the Hamming weight of a vector is the number of non-zero coordinates of this vector. Denote by $\Omega^{(t)}$ the set of elements of $\Omega = \Omega_1 \times \Omega_2 \times \ldots \times \Omega_k$ of the Hamming weight at most $t$.

We start by recovering the entries of the tensor that correspond to the indicies in $\Omega^{(1)}$.

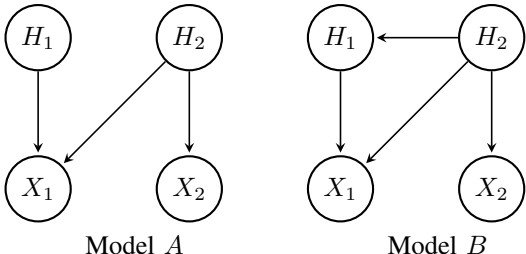

Model $A$          Model $B$

Figure 6: An example of the causal latent models that cannot be distinguished from observed data since Assumption 3.1 is violated

Let us pick an arbitrary mixture component $C$ that participates in the observed mixture model and let us put it in correspondence to $h = (0, 0, \ldots 0)$. We assign the probability of observing $C$ to the cell $J(0, 0, \ldots, 0)$.

Take any $i \in [m]$. Consider the set of $d(H_i)$ mixture components $\{C_{i,a} \mid a \in \Omega_i\}$, guaranteed by Lemma D.1, that have the same distribution as $C$ in coordinates $X \setminus \text{ch}(H_i)$ (here we take arbitrary indexing by $a$). Assign $C_{i,a}$ to the vector $h_{i,a} \in \Omega^{(1)}$ of Hamming weight 1, that has unique non-zero value $a$ in coordinate $i$. And let $J(h_{i,a})$ be the probability of observing $C_{i,a}$.

Next, we claim that the (valid) correspondence $\Omega \ni h \leftrightarrow i \in [K]$ for $h \in \Omega^{(t)}$ can be uniquely extended to the (valid) correspondence $\Omega \ni h \leftrightarrow i \in [K]$ for $h \in \Omega^{(t+1)}$ for any $t = 1, \ldots, m - 1$.

Indeed, let $h \in \Omega^{(t+1)}$ and let $i$ and $j$ be a pair of distinct non-zero coordinates of $h$. Let $h_i$ and $h_j$ be the vectors obtained by changing the $i$-th and $j$-th coordinates of $h$ to 0. Let $C_i$ and $C_j$ be the mixture components that correspond to $h_i$ and $h_j$.

Using Lemma D.1, for $s \in \{i, j\}$ we can find a set $M_u$ of $\dim(H_u)$ mixture components that are equally distributed with $C_s$ over $X \setminus \text{ne}_\Gamma(H_s)$. We put into correspondence with $h$ the unique component in the intersection of $M_i$ and $M_j$. We define $J(h)$ to be the probability of observing this component. $\qquad\square$

Next we show that our algorithm works in time that is almost linear in the output size (recall that $K \geq 2^m$ and $K$ is the size of the output).

**Observation D.4.** *The algorithm described in Theorem 5.4 works in $O((nm + \max_i k_i)K)$ time.*

*Proof.* First, the algorithm in Theorem 5.4 computes the equivalence classes of components that correspond to states of latent variables that differ just in the value of $H_j$. Having access to $\Gamma$ and $L$, computing these equivalence classes takes at most $O(nmK)$ time (for each of the $m$ hidden variables we need to compare vectors of values of $L$ of length $n$ for $K$ components).

Once these equivalence classes are computed, the algorithm in Theorem 5.4 sequentially fills in the joint probability table. If the entries with indices of Hamming weight $t$ are filled in, in order to determine the value of a cell with an index of hamming weight $t+1$, we explore at most $2 \max_{i \in [m]} k_i$ elements of the corresponding equivalence classes. Since eventually we explore all $K$ cells of the joint probability table, the total runtime of this phase is bounded by $O(\max_{i \in [m]} k_i)K$. $\qquad\square$

### D.4 Proof of Observation 5.5

Finally, we prove the impossibility claim in Observation 5.5.

*Proof of Observation 5.5.* We claim that if Assumption 3.1 is violated, then $\mathbb{P}(H)$ cannot be recovered and moreover $G$ is not identifiable. Consider a pair of models on Figure 6, where variables $H_1$ and $H_2$ are binary, i.e., they take values $\{0, 1\}$. Let $N_0, N_1, N_2, N_3$ and $N'_0, N'_1$ be independent Gaussian distributions with distinct means and variances.

Suppose that the observed distribution is equal to

$$(X_1, X_2) \sim \frac{1}{9} N_0 \otimes N_0' + \frac{2}{9} N_1 \otimes N_1' + \frac{2}{9} N_2 \otimes N_0' + \frac{4}{9} N_3 \otimes N_1' \qquad (14)$$

Now we show that this distribution can be realized by both models A and B.

1. *Consistency with A.* Let $H_1, H_2$ be independent random variables that take values $\{0, 1\}$ with probabilities $(1/3, 2/3)$.

$$(X_1, X_2) \sim \sum_{i \in \{0,1\}} \sum_{j \in \{0,1\}} \mathbb{P}(X_2 | H_1 = i, H_2 = j) \, \mathbb{P}(H_1 = i) \, \mathbb{P}(H_2 = j), \text{ where}$$

$$\mathbb{P}(X_1 | H_1 = 0, H_2 = 0) = N_0, \quad \mathbb{P}(X_1 | H_1 = 0, H_2 = 1) = N_1,$$
$$\mathbb{P}(X_1 | H_1 = 1, H_2 = 0) = N_2, \quad \mathbb{P}(X_1 | H_1 = 1, H_2 = 1) = N_3,$$
$$\mathbb{P}(X_2 | H_2 = 0) = N_0', \quad \mathbb{P}(X_2 | H_2 = 1) = N_1'$$

2. *Consistency with B.* Let $H_1, H_2$ be binary random variables with the following distribution

$$\begin{aligned} \mathbb{P}(H_2 = 0) = 1/3 \quad, \mathbb{P}(H_1 = 0 | H_2 = 0) = 1/3, \quad \mathbb{P}(H_1 = 1 | H_2 = 0) = 2/3, \\ \mathbb{P}(H_2 = 0) = 2/3, \quad \mathbb{P}(H_1 = 0 | H_2 = 1) = 2/3, \quad \mathbb{P}(H_1 = 1 | H_2 = 1) = 1/3 \end{aligned} \qquad (15)$$

Define components of the mixture distribution to be

$$(X_1, X_2) \sim \sum_{i \in \{0,1\}} \sum_{j \in \{0,1\}} \mathbb{P}(X_2 | H_1 = i, H_2 = j) \, \mathbb{P}(H_1 = i, H_2 = j), \text{ where}$$

$$\mathbb{P}(X_1 | H_1 = 0, H_2 = 0) = N_0, \quad \mathbb{P}(X_1 | H_1 = 0, H_2 = 1) = N_3,$$
$$\mathbb{P}(X_1 | H_1 = 1, H_2 = 0) = N_2, \quad \mathbb{P}(X_1 | H_1 = 1, H_2 = 1) = N_1,$$
$$\mathbb{P}(X_2 | H_2 = 0) = N_0', \quad \mathbb{P}(X_2 | H_2 = 1) = N_1'$$

Since both models $A$ and $B$ realize distribution $\mathbb{P}(X)$, we get that $G$ and $\mathbb{P}(H)$ are not identifiable. Observe that Assumption 3.1 is not satisfied for both $A$ and $B$, while Assumptions 2.2, 2.3 and 2.4 are satisfied for each of $A$ and $B$. $\qquad\square$

## E  Proof of Theorem 3.2

Finally, we collect our results into a proof of the main theorem.

*Proof of Theorem 3.2.* Suppose that Assumptions 2.2, 2.3 and 2.4 hold, then by Theorem 4.2(a), $\Gamma$ and $\dim(H_i)$, for all $i$, can be recovered from $\mathbb{P}(X)$. If additionally, the columns of the $|X| \times |H|$ adjacency matrix $A$ are linearly independent, then by Theorem 4.8 (see Corollary C.12, Theorem C.13 and Observation C.8), $\Gamma$ and $\dim(H_i)$, for all $i$, can be reconstructed efficiently in $O(n^4)$ time.

Now, suppose that Assumption 3.1 holds. We can extract the map $L$ from the MixOracle (by taking appropriate projections of component distributions). Therefore, since we have $\Gamma$, $\dim(H_i)$, $\{\pi(X, i)\}_{i \in [K]}$ and $L$, by Theorem 5.4 and Observation D.4, we can reconstruct $\mathbb{P}(H)$ efficiently. $\quad\square$

## F  Algorithms

In this section we describe the full pipeline[1] for learning $G$ from samples of the observed data $X$. As input we receive a set of samples and as output we return an estimated causal graph $G$ and a joint probability distribution over $H$. The pipeline consists of the following blocks:

(Step a) **Learning number of components.** Estimates the number of components for all subsets of observed variables of size at most 3.

  • Input: Samples from the distribution $\mathbb{P}(X)$

---

[1]The code used to run the experiments can be found at https://github.com/30bohdan/latent-dag

- Output: Estimated number of mixture components $k(S)$ in $\mathbb{P}(S)$ for all $S \subseteq X$, $|S| \le 3$.

(Step b) **Reconstruction of the bipartite graph.** Implements the algorithm of Theorem 4.8 for learning the bipartite causal graph $\Gamma$.

- Input: The number of mixture components $k(S)$ in $\mathbb{P}(S)$ for all $S \subseteq X$, $|S| \le 3$.
- Output: Estimated bipartite graph $\Gamma$ and sizes of the domains of hidden variables $\dim(H_i)$.

(Step c) **Learning the projection map $L$.**

- Input: Samples from the distribution $\mathbb{P}(X)$ and the numbers of components $k(X)$ and $k(X_i)$ for every $i \in [n]$.
- Output: Estimated projection map $L$.

(Step d) **Learning the distribution $\mathbb{P}(H)$.** In this step we implement the algorithm described in Theorem 5.4, see also Algorithm 1.

- Input: $L$, $\Gamma$ and $\dim(H_i)$ for all $i \in [m]$ and weights $\pi(X, j)$ of $k(X)$ mixture components.
- Output: Estimated joint probability table of $\mathbb{P}(H)$.

We take $L$, $\Gamma$ and $\dim(H_i)$ for all $i$ as an input and return the joint probability table for $\mathbb{P}(H)$ as an output.

(Step e) **Learning latent DAG $\Lambda$.** In this step we estimate the causal graph over latent variables.

- Input: The joint probability table of $\mathbb{P}(H)$.
- Output: Estimated causal graph $\Lambda$ over $H$.

In this paper, we prove theoretical guarantees for Steps (b) and (d), which invoke the mixture oracle MixOracle. Step (a) implements MixOracle, and Steps (c) and (e) are intermediate steps of the pipeline. As long as the oracle is correct, Step (c) is guaranteed to output the correct graph. The correctness of Step (e) depends on the structure learning algorithm used. A nice feature of our algorithm is its modularity, if a better algorithm is developed for one of the steps, it can be incorporated without influencing other parts.

Below we discuss various implementation details for these steps.

**Details of Step (a):** Our implementation of Step (a) uses the following strategy.

1. We estimate the upper bound $k_{max}$ on the number of components involved in the mixtures of single variables (this can be done using the silhouette score).

2. For every observed variable $X_i$ we train $K$-means clustering with $k = k_{max}$. After this, we perform agglomerative clustering for every $t \in [2, k_{max}]$, and record the silhouette score for every $t$. We pick 5 values of $t$ with the best silhouette score.

3. We use the divisibility condition to compute the sets $S_{X_i, X_j}$ of possible numbers of components we expect to see over the pairs of variables $X_i, X_j$. We use the best 5 predictions from the previous step for every variable $X_i$ and include the candidate for the number of components into $S_{X_i, X_j}$ if it is divisible by one of the top-5 candidates for $X_i$ and for $X_j$. This step is mainly needed for computational purposes in order to restrict the number of candidates for the number of components observed over the pairs of variables.

4. Next we learn the mixture of $k$ components for every $k \in S_{X_i, X_j}$ over the pairs $(X_i, X_j)$ of observed variables. Similarly as in 2., we train $K$-means for the largest candidate and perform agglomerative clustering after that.

5. We use divisibility and means voting (discussed in Sec. 6) to decide the best number of components for the single variables and the pairs of variables. In order to do this we make the predicted numbers of components for a pair $X_i$, $(X_i, X_j)$ to vote for each other if they satisfy the divisibility or means projection condition. We count the vote with the weight proportional to the silhouette score of the predicted number of components. For every $X_i$, and every pair $(X_i, X_j)$, we take the component with the largest amount of votes as our best prediction.

6. We use means of the components predicted for pairs of variables $(X_i, X_j)$ to estimate the locations of the means for the triples of observed variables. Instead of using $K$-means with the fresh start we initialize it with predicted locations. This improves the running time. We use $K$-means and silhouette score to predict the number of components for the triples of observed variables.

**Details of Step (b):** In this step we use Corollary 4.4, Eq. (6) and Lemma 4.6 to compute entries of the tensor $M_3$ using the output of Step (a). After this we apply Jennrich's algorithm to learn the components of the tensor. As discussed in Appendix C.3 this is sufficient to reconstruct $\Gamma$ and $\dim(H_i)$. In case Jennrich's algorithm did not successfully execute due to numerical issues, alternating least squares (ALS) was used as a failsafe. In this case, the number of hidden variables $m$ was used as input.[2]

**Details of Step (c):** We use $\Gamma$ and $\dim(H_i)$ to compute the number of components we expect to observe in $\mathbb{P}(X_i)$ for every observed variable $X_i$ and the number of components in the distribution $\mathbb{P}(X)$ over the entire set of observed variables. After this we use $K$-means to learn the components in the mixture distribution over every variable $X_i$ and over the entire set of observed variables. For every $i$, and for every mixture component of $\mathbb{P}(X)$, we project its mean into the subspace over which $X_i$ is defined. We use the closest in $L_2$ distance mean of the components in $\mathbb{P}(X_i)$ as a prediction for the projected component.

**Details of Step (d):** We implement the algorithm described in Theorem 5.4. See Algorithm 1 for details.

**Details of Step (e):** Once we obtain the estimated joint probability table, we run the Fast Greedy Equivalence Search [57] to learn the edges of the Latent graph $H$, where we used the Discrete BIC score. FGES returns a CPDAG by default, so some edges may be undirected. We accordingly report both the Structural Hamming Distance (SHD) and the Unoriented Correct Edges (UCE) as metrics for our experiments. We remark that this step may be improved by using other algorithms such as PC [69] or other scores, which is an interesting direction for future work.

## G Experiment details

**Data generation** For each experiment, the data generation process was as follows:

- $(m, n)$: Chosen from among $(1, 3), (2, 5), (3, 7), (3, 8), (4, 7), (4, 8)$ in the ratio $1 : 2 : 2 : 3 : 1 : 1$
- Domain sizes $|\Omega_i|$: Sampled from $\{2, 3, 4, 5, 6\}$. If $|\Omega| = |\Omega_1| \dots |\Omega_m| > 50$, we skip the experiment.
- $\mathbb{P}(H)$: Generated via the Markov property. For each variable $H_i$, conditioned on its parents $H_{\text{pa}(i)}$, a discrete distribution supported on $\Omega_i$ is chosen as follows: For each element $i$ in $\Omega_i$, a random integer $c_i$ is picked from $[1, 4]$ and distribution picks $i$ with probability proportional to $c_i$.
- $\Lambda$: Choose an arbitrary topological order uniformly at random and sample each directed edge independently with probability $0.6$.
- $\Gamma$: Sample each directed edge from $H$ to $X$ with probability $0.5$. Enforce assumption 3.1 and linear independence of the columns $a_j$ of the adjacency matrix $A$.
- Components: We generate Gaussian components for every $X_i$ in $\mathbb{R}^5$ with random means and covariances. We take the means of the components to be sampled uniformly at random from the unit sphere. We take random symmetric diagonally dominant covariance matrices with the largest eigenvalue being $0.01$. (Note that for 50 points on a unit 5-dimensional sphere, we expect to observe a pair of points at distance of the same order of magnitude).
- Samples: We generate samples from the mixture components generated on the previous step with probabilities defined by $\mathbb{P}(H)$.

---

[2]This can easily be avoided by running ALS for multiple values of $m$ and choosing the best fit. Since this issue arose in only a minority of cases, we did not implement this feature.

**Algorithm 1:** Learning $\mathbb{P}(H)$

---

**Input:**

- A bijective map $L : [k(X)] \to [k(X_1)] \times [k(X_2)] \times \ldots \times [k(X_n)]$;
- A bipartite graph $\Gamma$ between $X$ and $H$
- Values $\dim(H_i)$ for $i \in H$.
- Values $\mathbb{P}(Z = i)$ for $i \in [k(X)]$ (the probabilities of observing the mixture components)

**Output:** An $\dim(H_1) \times \ldots \times \dim(H_m)$ tensor such that $J \cong \mathbb{P}(H)$

```
// Phase 1:  use Lemma D.1 to compute the sets of components that
   correspond to a change in a single hidden variable
```
1   arrows = {}
2   **for** $H_i \in H$ **do**
3     S = $X \setminus ne_\Gamma(H_i)$
4     **for** $c_1, c_2 \in [k(X)]$ **do**
5       **if** $(L(c_2)_S == L(c_1)_S)$ *and* $c_1 \neq c_2$ **then**
6         arrows$[H_i][c_1]$.append$(c_2)$

```
// Phase 2:  initialize T "along the edges"
```
7   $A(0, \ldots 0) = 0, \quad T(0, \ldots 0) = \mathbb{P}(Z = 0)$
8   **for** $H_i \in H$ *and* $t \in \dim(H_i)$ **do**
9     $A(0, \ldots, t, \ldots 0) = arrows[H_i][0][t]$ // Note that an order does not matter
10   $J(0, \ldots, t, \ldots 0) = \mathbb{P}(Z = arrows[H_i][0][t])$

```
// Phase 3:  reconstruct all other entries of the tensor
```
11   $r = 1$
12   **while** $r < m$ **do**
13     **for** $ind \in \dim(H_1) \times \ldots \dim(H_r)$ **do**
14       **for** $j = r + 1, \ldots, m$ *and* $t \in \dim(H_t)$ **do**
15         Let $i$ be the smallest index at which $ind$ is non-zero.
16         Let $ind'$ be an index obtained from $ind$ by changing $j$-th entry from 0 to $t$
17         Let $ind''$ be obtained from $ind'$ by changing $i$-th entry to 0.
18         Let $x$ be the unique entry in the intersection of arrows$[H_i][A(ind'')]$ and
         arrows$[H_t][A(ind)]$.
19         $A(ind') = x$
20         $J(ind') = \mathbb{P}(Z = x)$

21   **return** $T$

---

We do not enforce minimum probability sizes or cluster sizes. As a result, the data generating process is likely to generate models which are extremely difficult to learn (e.g. if a randomly generated probability is very small, a mixture component will have few samples, which makes learning difficult). As a result, some random configurations may fail. We ran a total of 724 experiments; out of these, 8.3% failed in the oracle learning phase and another 8.8% failed to produce a graph because of very high domain sizes or unfeasible $L$. In the cases when the Jennrich algorithm failed due to numerical issues, this was caught and replaced with ALS for practical purposes as described in Step (b) above. These errors are conveniently caught during runtime and can be attributed to either the data generation process or the finite sample size as described above. Fig. 3 reports the metrics for the remaining 600 experiments: 300 experiments each for $N = 10000$ samples and $N = 15000$ samples. The experiments were run on a single node of an internal cluster.

**Experiments with smaller sample size.** The number of samples in the experiments discussed above is chosen so that every cluster component has approximately 20 samples. We also explored the behaviour of our algorithms when the number of samples is much smaller. We ran a total of 136 experiments for $N = 1000$ samples, with $(m, n)$ chosen from $(1, 3), (2, 5), (3, 7), (4, 7), (3, 8)$ in proportion $1 : 2 : 1 : 1 : 1$. Out of these, 4.4% failed in the oracle learning phase and another 8.8% failed to produce a graph because of very high domain sizes or unfeasible $L$. Furthermore, out of

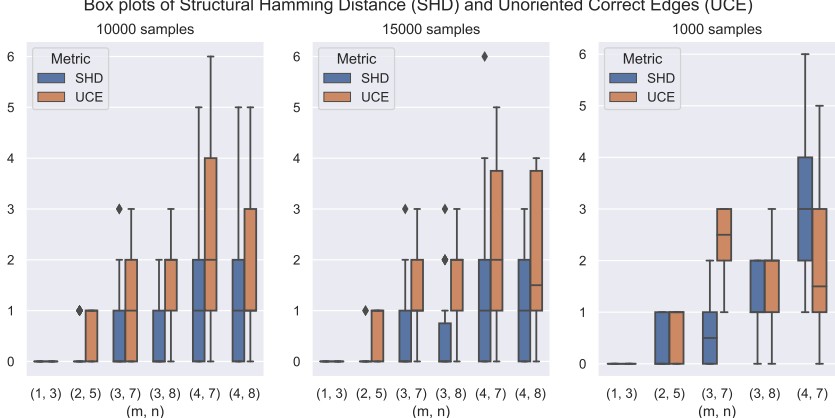

Figure 7: Average Structural Hamming distance for recovery of $G$, where $m = |H|$ and $n = |X|$.

all failures, $25\%$ happen for $(m, n) = (4, 7)$ and other $37.5\%$ happen for $(m, n) = (3, 8)$. We report the metrics on Fig. 7.

We mention, that with $N = 1000$ samples, we were able to recover $H$ and $\Omega$ even in the cases when several latent states had fewer than five observations. Also, for comparison, to give an example where we were not able to recover $H$ and $\Omega$ exactly: the mixture model had 48 components with $1, 2, 2, 3, 3, 5, 5, 5, 6 \ldots, 53, 55$ samples per component. This is clearly an extremely challenging setup: Some states had only a few observations and the true number of components is unknown to the procedure.

**Choice of parameters for learning** $\Lambda$. Once we have recovered the estimated joint probability table of $H$, to learn $\Lambda$, we use the Fast Greedy Equivalence Search algorithm [57] with the Discrete BIC score. We use the PyCausal library [77]. We used the default parameters (no hyperparameter tuning) and in particular, we did not assume faithfulness.

**Approximate Runtime**    The average runtimes for each experiment are in the following table.

Table 1: Average runtime in seconds

| (m, n) | 10000 samples | 15000 samples |
|--------|---------------|---------------|
| (1, 3) | 30.64 s | 53.06 s |
| (2, 5) | 89.03 s | 148.81 s |
| (3, 7) | 288.25 s | 385.27 s |
| (3, 8) | 320.25 s | 616.86 s |
| (4, 7) | 297.32 s | 400.04 s |
| (4, 8) | 361.28 s | 604.14 s |

**Average number of edges**    For our experiments, the average total number of edges in $\Lambda, \Gamma$ (also known as NNZ of $G$) are in the following table.

Table 2: Average number of edges for different settings

| (m, n) | Number of Samples | Average number of edges in $G = (\Lambda, \Gamma)$ |
|--------|-------------------|---------------------------------------------------|
| (1, 3) | 10000 | 3.0 |
| (1, 3) | 15000 | 3.0 |
| (1, 3) | 1000 | 3.0 |
| (2, 5) | 10000 | 7.15 |
| (2, 5) | 15000 | 6.95 |
| (2, 5) | 1000 | 6.98 |
| (3, 7) | 10000 | 13.52 |
| (3, 7) | 15000 | 13.2 |
| (3, 7) | 1000 | 13.7 |
| (3, 8) | 10000 | 15.27 |
| (3, 8) | 15000 | 15.16 |
| (3, 8) | 1000 | 15.3 |
| (4, 7) | 10000 | 17.43 |
| (4, 7) | 15000 | 18.17 |
| (4, 7) | 1000 | 18.35 |
| (4, 8) | 10000 | 19.87 |
| (4, 8) | 15000 | 20.13 |

**Scatter plots**    The scatter plots for the Structural Hamming distance (SHD) versus the total number of edges $|E(G)|$ in $G$ and that of the unoriented correct edges (UCE) vs $|E(G)|$ is given in Fig. 8.

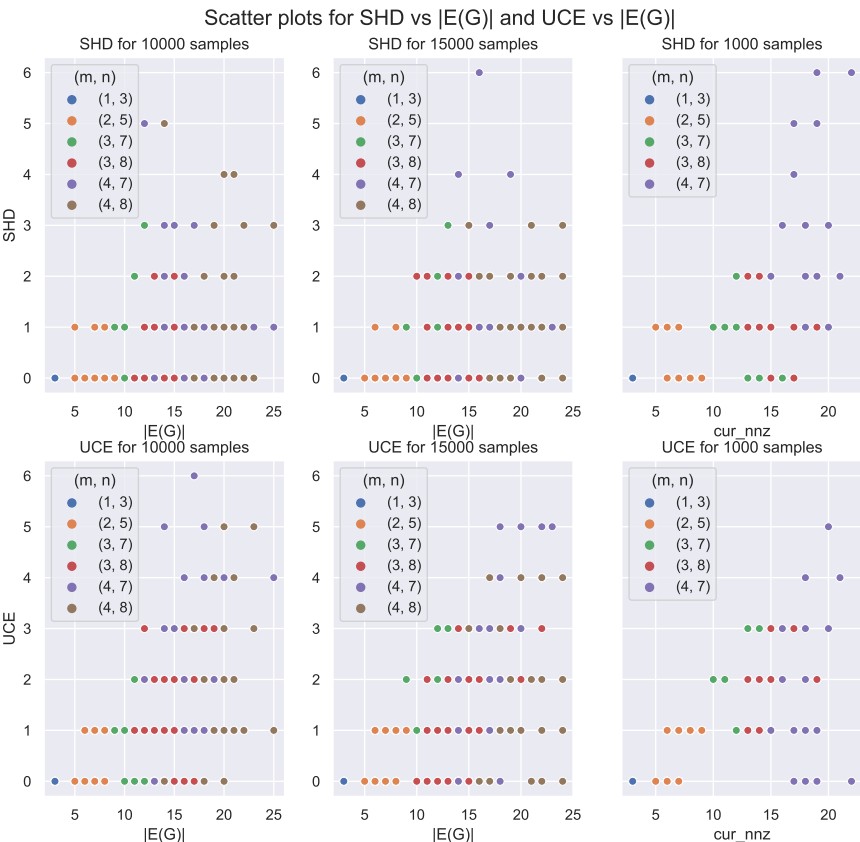

Figure 8: Scatterplots where $m = |H|$ and $n = |X|$.