# OpenReview forum: "Learning latent causal graphs via mixture oracles"
_NeurIPS.cc/2021/Conference — NeurIPS 2021 Poster_

### Official Review · Reviewer_1Np2 · 2021-06-28

**Rating:** 6
**Confidence:** 4

**Summary:**

This paper considers learning discrete latent variables (the causes) and their dependencies, given observed variables. It is assumed that there is a DAG between the hidden nodes and a bipartite graph with edges from the hidden nodes to the obseved nodes, and no edges between the observed nodes. The idea is that a distribution of each subset of observed variables is described by a mixture model where each component corresponds to an assignment of values to  hidden variables which are the causes of these observed variables. The paper then presents proofs that from such mixture models (leaned in practice by k-means in the article), the distribution and the graphs related to the hidden nodes can be recovered.

**Limitations And Societal Impact:**

Yes.

**Main Review:**

PROS:

-Straightforward and solid idea

-Solid mathematical presentation

CONS:

-Seems like time has run out while writing the article. I was expecting a convincing demonstration of the usefulness of the idea with some real-world data, but there was not even an attempt for such. Furthermore, the article had no Discussion/Conclusion.

SUMMARY: this paper introduced a sensible idea for causal discovery in the latent space, with explicit assumptions and formal proofs of identifiability, which I liked. However, the usefulness of the idea should have been demonstrated in a realistic use-case.


**Time Spent Reviewing:**

3

---

> ### Author Response · Authors · 2021-08-10
> **Response to Reviewer 1Np2**
>
> We thank the reviewer for their comments.
>
> In this paper we are dealing with a challenging problem of non-parametric causal discovery with discrete latent variables. Most of the prior work assumes linear models or known structure. In this sense, our results are very strong and greatly expand the scope of prior work. As such, we would like to emphasize that this is a primarily theory paper which proves rigorous necessary and sufficient conditions for identifiability and gives first such theoretical guarantees for the general measurement model we considered.  It is common for theory papers to have no experiments (real or simulated) at all.
>
> In contrast, we have implemented our algorithm and *included detailed simulations*, which we believe greatly enhances our contributions. The fact that our theoretical algorithm indeed translates into something practical---and works!---is surprising. We like to say that our approach works “surprisingly well” in practice.
>
> Finally, we would like to point out that in real world data, there is no ground truth, so any such evaluation would be very tricky. Since our focus have been on exact recovery and identifiability, it is not clear what such experiments would add to the rigorous theory we have developed (we certainly agree that *without* theoretical results, real world experiments are crucial to validating the approach).
>
> That being said, we are eager to explore real world datasets and improving the efficiency of our overall approach. With the extra space allowed in the camera ready, we will discuss these points, including potential real world applications.

---

> > ### Comment · Reviewer_1Np2 · 2021-09-02
> > **Thank you for your comments**
> >
> > Thanks for your comments! After discussing with the other reviewers, I agree that the article makes an interesting contribution in showing how to reduce the problem of latent causal graph learning into that of estimating a mixture oracle. While the paper could have been strengthened by more convincing experiments, this did not seem to bother the other reviewers as much. I support the idea of adding a corollary after Theorem 1 that includes the "examples of identifiable models", along with pointers to the corresponding identifiability results in the literature, suggested by reviewer mBxt. With this change I’m willing to be on the side of acceptance of this article.

---

> > > ### Author Response · Authors · 2021-09-02
> > > **Thanks!**
> > >
> > > Thank you for your response and willingness to change your score! We appreciate the concrete suggestion to add specific examples of identifiable mixture models after Theorem 1, along with the many other helpful comments from you and the other reviewers. We will be sure to carefully incorporate these changes into the final version.

---

### Official Review · Reviewer_9wNn · 2021-07-14

**Rating:** 7
**Confidence:** 3

**Summary:**

This is a paper that focuses on causal discovery especially focusing on the causal relation among discrete latent variables.
The basic idea is that by mixture oracles, one can locate the latent variables and recovery their distribution.
The main contribution is the model identification theory, including bipartite graph, latent distribution $P(H)$.

**Limitations And Societal Impact:**

Yes.

**Main Review:**

The paper considers a challenging problem of learning causal graphs over discrete latent variables.  The key point is how to determine the number of hidden variables and their corresponding indicators.\
The authors provide a  new view for identifying a discrete latent model (recover the bipartite graph by learning the parameters of the mixture model ).\
The theoretical contributions of this paper are interesting, and the experiments seem promising.

Some concerns:\
-The algorithm's complexity is a cause for concern, especially when the graph has a large number of variables. This is because the algorithm involves the enumeration of all subsets that the cardinality is less than or equal to 3.\
-The algorithm is evaluated with a large sample size in your experiments. It would have been more convincing with a small sample size.  I am worried about the usefulness of the algorithm.\
-It would be good if you show the limitations of your method in the conclusion.

-------After rebuttal-------

Thanks for the clarifications and for addressing my concerns. Based on the author's response and other reviews, my score remains unchanged.

**Time Spent Reviewing:**

6 hours

---

> ### Author Response · Authors · 2021-08-10
> **Response to Reviewer 9wNn**
>
> We thank the reviewer for their review and positive feedback.
>
> **Sample size:** The large sample size in our experiments was caused by the fact that the total number of latent states is as large as 50 and we intentionally did not enforce cluster probabilities to be balanced and clusters to be separated in order to make the settings more challenging.
> The sample sizes we tested were such that each latent state was observed at least ~20 times.
> Nonetheless, we agree with the reviewer that understanding lower limits of the sample complexity for our algorithm is important and we apologize for not reporting this in our original submission. We ran our code with 10x fewer samples for several settings of the parameters and we report the results below.
>
> As in the paper, let n denote the number of observed variables, m denote the number of latent variables, D denote the largest domain size of a single discrete observed variable.
> We ran our algorithm _without any modifications, with the same random seed_ for 1000 samples (compared to 10000 samples or more in the paper) for random DAGs with (m, n, D) = (2, 5, 3), (3, 7, 4), and (3, 8, 3). We ran 71 experiments in total, and in 84.5% of the experiments we recovered H and $\Omega$ *exactly* (20 exact recoveries for each setup), with only 1 or 2 errors in recovering $G$. In the cases when H and $\Omega$ were recovered exactly, the average SHD is 0.5 for (m, n, D) = (2, 5, 3) and 1.0 for (m, n, D) = (3, 7, 4) and 1.2 for (m, n, D) =(3, 8, 3) (maximal SHD is 1, 2 and 2, respectively). In all these experiments, every edge present in the true model was recovered correctly and positive SHD is only due to superfluous edges.
>
> With this sample size, we were able to recover H and $\Omega$ even in the cases when several latent states had <5 observations! Also, for comparison, to give an example where we were not able to recover H and $\Omega$ exactly: the model had 48 components with 1, 2, 2, 3, 3, 5, 5, 5, 6 …, 53, 55  samples per component. This is clearly an extremely challenging setup: Some states had only a few observations and the true number of components is unknown to the procedure.
>
> We will include these experiments with smaller sample size in the final version of the paper.
>
>
> **Regarding runtimes:** We believe that the running time can be improved by considering only subsets of size at most 2, or by using heuristic algorithms, and we are certainly interested in this as a subject for future work. That said, we mention that learning even very small instances of  DAGs with latent structure is known to be notoriously difficult. We hope to make these compelling theoretical results more practical in future work, and absolutely plan to discuss this in the final version of the paper.
>
> We are happy to discuss these limitations in more detail in the final version of the paper.

---

### Official Review · Reviewer_mBxt · 2021-07-16

**Rating:** 6
**Confidence:** 4

**Summary:**

The authors identify a set of assumptions under which it is possible to recover the joint distribution over the latent variables. My understanding is that they extend NMF to use tensor factorization and under suitable assumptions show that the tensor factorization solution is the unique solution - up to relabeling. I however have several issues parsing some of the results. I would appreciate feedback from the authors to help clarify the points below.


**Limitations And Societal Impact:**

No. Limitations could be discussed in more detail in relation to my comments above.

**Main Review:**

The authors should present their result in light of some of the missed existing work such as pLSA or LDA. In general, the non-negative matrix factorization literature provides us with certain impossibility results which is only somewhat discussed in Section 3 (anchor words).

I believe the model the authors assume here is called the measurement model - where there are no edges between the observed variables and no edges from observed variables to latents.

Essentially authors only assume confounders and no latent mediators.

Additional assumptions:
1. no pair of distinct latents are adjacent to the same set of observed variables.

2. this is a complicated assumption and is a bit hard to judge. Essentially that there is no smaller DAG with same assumptions according to which the distribution is Markov.

3. strictly positive latent dist. and unique conditional distributions for every subset of target variables.

4. There is a 4th hidden assumption that the mixture oracle is available. Essentially, mixture oracle does all the heavy-lifting by identifying not only the number of mixture components but also the mixed distributions and the mixture weights.

5. There is a 5th assumption that appears in Section 3 which says no neighborhood of any two latent nodes have subset relation.

"This assumption is weaker than the common “anchor words" assumption from the topic modeling literature. "
Could you elaborate on why this is?

Authors claim their condition for recoverability is weaker than anchor node condition. Anchor node condition is one sufficient condition for ensuring a unique NMF. Does this mean the proposed method provides a weaker condition for unique recoverability? I don't believe this is the case but I think this needs to be discussed and clarified in the main text.

On that note, I believe one can easily construct NMF examples, which correspond to mixture models, where the solution is not unique but all the required assumptions are satisfied. This relates to the uniqueness of (2).

Please comment on why the representation in (2) has to be unique. For example, I can write p(x)=[1/4,3/4] as multiple mixtures \sum p(h)p(x|h), essentially any a, b, c, d, e would do s.t a[b,c]+(1-a)[d,e]=[1/4,3/4]. E.g. take 0.5[0,1]+0.5[1/2,1/2] or take 0.25[1,0]+0.75[0,1]. In either case, p(x|h) are distinct (condition in Assumption 2.3) whereas the other representation fits just as well.

I am also not sure if Observation 2.6 is accurate. It would be great to have a more formal and rigorous proof. The above discussion and example seems to constitute against the argument in the current proof which implicitly seems to rely on uniqueness of (2).

Fig. 2 is a good example and it demonstrates that assumptions essentially imply each mixture component can be observed in the joint distribution of the observed variables. In that sense, I don't fully understand how it is weaker than anchor node assumption.

It is very hard to assess the validity of all the assumptions without real data experiments.

Unless I am misunderstanding something, tensor decomposition is not poly time unless the number of components is a constant. But a generic poly-time claim is made in the introduction and the detailed discussion does not appear in the main text but only in the appendix.

Figure 1 is not very useful since it is disconnected from the discussion. I recommend explaining in which of these graphs the assumptions hold. I believe it should hold in all of them.

After Rebuttal:
Thank you to the authors for engaging in a very useful discussion. Authors made clear where their contribution lies and that their identifiability result heavily relies on the existence of a mixture oracle. They provided examples of such oracles in certain settings. In light of the discussions, I will increase my score.

**Time Spent Reviewing:**

4

---

> ### Author Response · Authors · 2021-08-10
> **Response to Reviewer mBxt**
>
> We are grateful to the reviewer for thorough review and detailed feedback.
>
> > _My understanding is that they extend NMF to use tensor factorization_
>
> While pLSA, LDA and NMF are related to the problem considered in the paper, our results are neither a generalization, nor a corollary of the existing work in this literature. We will add more discussion about the relation between these problems in the final version, and will be happy to provide relevant references and comparisons. Please see below for these details.
>
> In more detail: Although the problems are clearly related (identifying latent structure), our setup is entirely distinct from the setup in LDA, and bears very little resemblance to the NMF approach taken in this literature. For example, one standard approach to LDA is to form a term-by-document matrix M, which is to be approximated by a matrix factorization of the form M=AW, where A collects the topic distributions and W collects distributions over topics. NMF can then be used to factor the matrix of second moments $MM^T$ (co-occurrences of words) into the matrices of topics’ distributions over words and a matrix of topics’ co-occurrences.
>
> This needs to be contrasted with our approach, which doesn’t even rely on higher order moments (i.e. a moment tensor), which _would_ be a natural generalization of the NMF approach. Instead, our tensor is completely different: It is a carefully constructed tensor whose entries only depend on the numbers of components in the mixture distribution and not on their weights or distributions of individual components. _After performing this tensor decomposition_, we then reconstruct the actual latent distribution in a completely different way (see Section 5, Appendix D, Algorithm 1 in the supplement).
>
> In summary, to the best of our knowledge, there is not much formal relationship between NMF and our approach. Nonetheless, we agree that the LDA/NMF literature warrants further discussion and comparison, which we are happy to include in the final version.
>
> > _The authors should present their result in light of some of the missed existing work such as pLSA or LDA. In general, the non-negative matrix factorization literature provides us with certain impossibility results which is only somewhat discussed in Section 3 (anchor words)._
>
> We are happy to add more discussion about the relation between pLSA/LDA and our problem in the final version, and will be happy to provide relevant references and comparisons. For the reasons explained above, our problem is quite a bit different, but we agree that this related work deserves a more prominent discussion which was unfortunately overlooked (largely due to the differences outlined above).
>
> We would also like to emphasize that due to these differences, the complete set of our assumptions is not directly comparable to the assumptions made in the pLSA/LDA literature. This is precisely why the anchor words assumption was only mentioned after Assumption 3.1, which is naturally comparable to the anchor words assumption in isolation. The rest of our assumptions are particular to our setup (although we are happy to hear about other possible connections if the reviewer is aware of any).
>
> Finally, we note that our results nicely complement the impossibility results from this related work since we also show that all assumptions we make are actually *necessary* (L130, L139, App. B, App. D4) for identifiability. If any of these assumptions are dropped, our latent causal model problem becomes non-identifiable unless alternative assumptions are made.
>
> > _Please comment on why the representation in (2) has to be unique._
>
> Every representation (1) canonically induces a representation (2) (by ignoring all the latent structure in H). Therefore, identifiability of (2) is a *necessary condition* for identifiability of (1).
> Our paper shows that under mild additional assumptions, identifiability of (2) is sufficient to identify $\Omega$, H, P(H) and representation (1) and solve the challenging task of recovering the latent causal structure. We assume access to the representation (2) via a Mixture Oracle, for which we describe a flexible framework and provide an example of implementation.
>
> As the reviewer has correctly pointed out, mixture models such as (2) need not be identifiable in general. As we have discussed at L94-105, there is a long line of work on this problem with positive identifiability results, including in fully nonparametric settings. This is important to confirm that the uniqueness of (2) is not an issue, and we appreciate the reviewer’s close attention to this issue. We have cited 6 such example papers (L99-101) relevant to our work, and are happy to add additional citations on this point since the literature on this problem is quite vast.
>
> > _this is a complicated assumption and is a bit hard to judge._
>
> Regarding Assumption 2.2: It appears there is an unfortunate misunderstanding here, which is partially on us for not describing this assumption more clearly in the submission. Assumption 2.2.2, although correct as stated, would perhaps be more easily interpreted if it reads “$G’$ is obtained from $G$ by _splitting_ a hidden variable" (note that $G’$ and $G$ have swapped places here). In other words, this assumption says that no hidden vertex can be _split_ into two vertices without introducing a pair of twins, while still satisfying the Markov property.
>
> Thus, Assumption 2.2 says that there is no *larger* graph (without twins) that is Markov with respect to the given distribution (meaning we are interested in recovering as much structure as possible). As discussed in L130, L139 this assumption is necessary for identifiability.
>
> > _Authors claim their condition for recoverability is weaker than anchor node condition._
>
> We apologize if this caused any confusion. To clarify: Our claim (L191) is just that Assumption 3.1 alone is weaker than the anchor word assumption, which is clear from the definition. This is also illustrated by Fig 2 (L278), where the anchor word assumption is violated for Fig 2 as there is no observed variable that is adjacent only to $H_1$.
>
> > _Unless I am misunderstanding something, tensor decomposition is not poly time unless the number of components is a constant._
>
> Without *any* assumptions, it is certainly true that tensor decomposition is NP-hard [3]. In our setting, however, we are able to use Jennrich’s algorithm (Theorem 4.8), which is provably poly time for any number of components [4].
>
> > _I am also not sure if Observation 2.6 is accurate._
>
> This observation is a direct corollary of Assumption 2.3 and identifiability of (2), as discussed in the proof. Moreover, Observation 2.6 is absolutely essential to our proof technique and our overall approach, and there is no chance that the algorithm implemented in Sections 6-7 would succeed if Observation 2.6 was not accurate.
>
> Thank you again for your thorough review, we will be happy to answer any further questions.
>
> [1] Blei, D., Lafferty, J. A correlated topic model of science. Ann. Appl. Stat. (2007), 17–35.
>
> [2] Anandkumar, Animashree, et al. "Learning linear bayesian networks with latent variables." International Conference on Machine Learning. PMLR, 2013.
>
> [3] J. H˚astad. 1990. Tensor rank is NP-complete. J. Algor. 11, 4, 644–654.
>
> [4] Moitra, Ankur. "Algorithmic aspects of machine learning." (2014).

---

> > ### Comment · Reviewer_mBxt · 2021-08-20
> > **rebuttal responses**
> >
> > Hello. Thank you very much for your constructive comments and detailed explanations.
> >
> > I have re-read the paper in light of the authors' responses. I am realizing that the main source of my confusion was the identifiability claim which is used as the punchline throughout the paper despite the fact that most of the identifiability issue is offloaded to a Mixture oracle that is assumed to be available. I strongly believe the notion of "identifiability" shouldn't require access to an oracle that typically is not available, but the assumptions must be in a way that we can start from data and directly go to the answer. This is not currently the case. I recommend the authors to reconsider the narrative and paraphrase the identifiability claim or add the extra assumptions to their assumptions such that the Mixture oracle would be available in order to paint a more complete picture. (I wasn't able to spot such an assumption. Only a k-means based implementation is mentioned as far as I can see).
> >
> > Having said that, the paper demonstrates that EVEN IF we assume access to such an oracle, there are still several challenges that need to be addressed and this is where the paper's contribution lies. Particularly, now I better understand the use of tensor methods which nicely leads the way for Theorem 4.8.

---

> > > ### Author Response · Authors · 2021-08-22
> > > **Mixture identifiability is very well-studied**
> > >
> > > > _I strongly believe the notion of "identifiability" shouldn't require access to an oracle that typically is not available, but the assumptions must be in a way that we can start from data and directly go to the answer._
> > >
> > > We apologize for any confusion on this: To clarify, **as long as the mixture model (2) is identifiable, this mixture oracle exists** (see L163-164, also Remark 5 where it is clarified that the full power of MixOracle is not really needed). This is not a strong assumption: **Identifiability in mixture models is an exceptionally well-studied problem, with a deep literature going back over 50 years** (please see our discussion at L94-105, where we have cited 6 such example papers @ L99-101). In other words, **in many practical settings an efficient mixture oracle is indeed available** (see below). We completely agree that identifiability shouldn’t require access to an unrealistic oracle; fortunately in our case such an oracle is not at all unrealistic and indeed very well-studied.
> > >
> > > Examples of identifiable mixture models include (see L94-101 for sample references):
> > > - Fully nonparametric mixtures that are separated in TV distance (this **does not** imply spatial or mean separation)
> > > - Product mixtures
> > > - Gaussian mixtures
> > > - Exponential family mixtures
> > > - etc.
> > >
> > > Any of these examples work within our framework, and yield a concrete implementation of a corresponding mixture oracle. Evidently there are many possible choices, each with its own conditions under which the mixture model is identifiable. A key aspect of our approach is that it does not rely on any *particular* mixture identifiability assumption, so there would be significant loss of generality in restricting attention to some special case. This is why we have phrased everything in terms of the mixture oracle.
> > >
> > > This in fact explains our choice of framing in terms of a mixture oracle: _This is the most well-studied and well-known aspect of the problem._ By contrast, the reduction of graph identifiability to mixture identifiability is novel and one of our main contributions, and we think an exciting development in the literature on learning causal graphs with latent variables.
> > >
> > > We hope this completely addresses your remaining concerns, and will be happy to provide additional clarification on these points in the final version.

---

> > > > ### Comment · Reviewer_mBxt · 2021-09-01
> > > > **response to rebuttal response**
> > > >
> > > > Hi. Thank you for your response. I skimmed the paper one more time in light of your pointers and in hindsight it is clarified several times that mixture oracle is assumed. Based on this, I will increase my score as the authors made it sufficiently clear where their contribution lies.
> > > >
> > > > I recommend adding a corollary after Theorem 1 that includes the "examples of identifiable models" you have in your response, along with pointers to the corresponding identifiability results in the literature. Something that reads like "Corollary 1: Suppose the data is generated either from i) .. ~\cite{.} or ii)..~\cite{.} then the causal graph is identifiable."
> > > >
> > > > This will draw a complete identifiability picture for the readers and also let them know which papers to look at to understand your identifiability result from start to finish. Thank you again for your engagement in the rebuttal.

---

> > > > > ### Author Response · Authors · 2021-09-02
> > > > > **Thank you!**
> > > > >
> > > > > We are happy that we were able to clarify this, and appreciate your time and understanding on this. Your suggestion to add specific examples after Theorem 1 is well-taken, and agree that this will help to present a complete picture. We will be sure to incorporate your suggestions (along with the other suggestions from the other reviewers) into the final version.

---

### Official Review · Reviewer_rWbo · 2021-07-25

**Rating:** 7
**Confidence:** 4

**Summary:**

This paper proposes a method for recovering a distribution over a set of discrete latent random variables, given observations from a set of observed children of these variables. The method additionally recovers the set of edges between latent and observed variables. The method is written in terms of a “mixture oracle”, i.e., a subroutine which returns mixture weights and components for the distribution over any subset of observed variables. Empirical results show that the method is effective at recovering the underlying causal graph.

**Limitations And Societal Impact:**

The authors adequately address the potential negative societal impact of their work (and the broader line of work on recovering latent variables) with respect to privacy. The paper takes a “white hat” perspective: it is good to know when certain latent variables can be recovered, so as to be aware of possible violations of privacy, and this line of work may offer enough insight to design systems which are provably immune to privacy attacks.

It would be beneficial to the authors to address the limitations of the work more explicitly, e.g. giving settings where their assumptions don’t hold (e.g., images as mentioned above) and where discrete random variables are a poor model. In particular, even though discretizing a continuous variable might allow the method to theoretically be applicable, the required alphabet size might make the methods too computationally expensive or statistically inefficient to be useful in practice.


**Main Review:**

Originality
--------------
The task considered, estimation of a latent causal graph, has been previously studied (Silva et al., 2006, Xie et al., 2020) and is of great interest to many in the causal inference community. The method/setting is new, and this novelty is made clear throughout the paper: most other works have relied on linear models, whereas this work considers the discrete setting. Additionally, other works have had more stringent conditions on the structure of the graph relating the latent and observed variables. The prior works are adequately discussed, and I appreciate that the authors mention a few separate lines of work, including causal feature learning, interaction modeling, and computer vision. However, for completeness, the authors may wish to add the following citations when discussing the most directly related line of work, i.e., learning latent causal structure:
- Cai et al., 2019: Triad constraints for learning causal structure of latent variables
- Kummerfeld and Ramsey, 2016: Causal clustering for 1-factor measurement models

Quality
----------
For the most part, the submission is theoretically sound, however, I have only skimmed the proofs in the appendix. There was one point of confusion for me in the main text. Assumption 2.2 requires that the same distribution cannot be Markov to any DAG formed by merging a pair of hidden variables in the true DAG. However, it seems to me that this would always be possible; for example, consider the graph in Figure 1b of the paper. Merging H1 and H2 would give a new DAG which satisfies Assumption 2.1 (no twins), and the distribution over (H1, H2, X1, X2) would still be Markov with respect to this new DAG (where now we replace H1 and H2 with a single variable H=(H1, H2)). In particular, the only implied conditional independence in the new model is that X1 and X2 are conditionally independent given H, which still holds. It seems I am misunderstanding the minimality assumption, since the authors claim that the graphs in Example B.2 satisfy this assumption. Please clarify in the author feedback.

The piece of work is complete by the standards of a conference: the authors not only provide an identifiability result, but offer computational complexity results for setting where an efficient algorithm is possible. There is definitely still interesting future work on this exact setting, including improving the subroutines used for estimating the mixture oracle, and a thorough statistical analysis, along with theoretical results for how to “vote” on the number of marginal components (mentioned in Section 6 with a very helpful example).

Clarity
----------
The submission is very clearly written, with only two points of confusion/poor wording. The first point regards Assumption 2.1, already mentioned. The second point is minor: the authors use phrases such as “little is known regarding structure learning … with latent variables”. The authors are correct when we restrict to structure learning *over* latent variables, i.e., when the interesting causal structure is mostly over the latent variables. However, structure learning *in the presence* of latent variables is a very rich field of study in causal structure learning, including work on learning maximal ancestral graphs (Richardson and Spirtes, “Ancestral Graph Markov Models”, 2002) and work on learning DAGs in the presence of pervasive confounders (Frot et al. “Robust causal structure learning with some hidden variables, 2019). The language of “with latent variables” is used in many places throughout the paper, and it would be an improvement to be more precise.

Significance
-----------------
The results are an important contribution to the somewhat nascent, but widely anticipated, area of learning latent causal structure. As mentioned, the paper shows an optimistic result about how “signatures” of latent variables can be used to recover the distribution over the latent variables themselves. It is likely that in the discrete setting, the building blocks of their methods will be used in future works and the details of the method will be built upon/improved. It is also likely that some algebraic or information theoretic characterization of the observations used here in the discrete setting could carry on to general (i.e., continuous and nonlinear) settings. On the less positive side, I should note that in practice, the actual setting - discrete random variables, and conditions on the relations between observed and latent variables - is likely too restrictive for immediate use in applications such as recovering causal structure from images, where every latent variable is likely to be a parent of every observed variable (pixel).

**Time Spent Reviewing:**

4

---

> ### Author Response · Authors · 2021-08-10
> **Response to Reviewer rWbo**
>
> We thank the reviewer for the detailed review and positive feedback.
>
> **Regarding Assumption 2.2:** It appears there is an unfortunate misunderstanding here, which is partially on us for not describing this assumption more clearly in the submission. Assumption 2.2.2, although correct as stated, would perhaps be more easily interpreted if it reads “$G’$ is obtained from $G$ by _splitting_ a hidden variable" (note that $G’$ and $G$ have swapped places here). In other words, this assumption says that no hidden vertex can be _split_ into two vertices without introducing a pair of twins, while still satisfying the Markov property. Merging is indeed always possible (L130-131). Assumption 2.2 is also necessary (L139-140) for identifiability (otherwise, as the reviewer has noted, latent variables can be created or destroyed without changing the observed distribution $P(X)$).
>
> **Structure learning over the latents:** We appreciate the reviewer pointing out the important distinction between “structure learning in the presence of latent variables” vs “structure learning over the latent variables”, and regret that this was not made more clear in our submission. We indeed should have been more careful here and we will be happy to fix this in the final version. We will also be sure to cite relevant work (in addition to the papers we have already cited, L21-24) when we make this fix, including the two references mentioned by the reviewer.
>
> **Previous work:** We are also happy to hear the reviewer is satisfied with our discussion of previous work, and will be more than happy to add references to the suggested papers, which we agree are very relevant and should be included.
>
> **Limitations:** We will be happy to discuss the limitations of our work in more detail in the final version of the paper. Our view on limitations and future directions for our work is very close to the reviewer’s, including potential applications to computer vision. Sorting out how to relax our conditions on the relation between observed and latent variables is a very important direction for future work that we are eager to explore.

---

> > ### Comment · Reviewer_rWbo · 2021-08-10
> > **Thank you for the clarifications!**
> >
> > Thank you for the clarification regarding Assumption 2.2 - this makes a lot more sense. I appreciate that the authors have embraced the feedback on the other points.

---

### Decision · Program_Chairs · 2021-09-27

**Decision:**

Accept (Poster)

**Comment:**

There has been ampled discussion between reviewers and authors. The authors are encourage to leverage the elements in this exchange to improve the paper.

In particular, a key conlcusion of the discussion between reviewers is that given the key role of a Mixture Oracle in this work, all reviewers agree that it would be really important to add a paragraph describing existing identifiability results for mixture oracles, as well as a corollary that gives an exact set of assumptions under which the algorithm is consistent.